# TIGAR deficiency enhances skeletal muscle thermogenesis by increasing neuromuscular junction cholinergic signaling

Yan Tang[1], Haihong Zong[1], Hyokjoon Kwon[1], Yunping Qiu[1], Jacob B Pessin[1], Licheng Wu[1], Katherine A Buddo[2], Ilya Boykov[2], Cameron A Schmidt[2], Chien-Te Lin[2], P Darrell Neufer[2], Gary J Schwartz[1,3,4], Irwin J Kurland[1,4], Jeffrey E Pessin[1,4,5]*

[1]Departments of Medicine, Albert Einstein College of Medicine, Bronx, United States; [2]East Carolina Diabetes and Obesity Institute and the Department of Physiology, Brody School of Medicine East Carolina University, Greenville, United States; [3]Departments of Neuroscience, Albert Einstein College of Medicine, Bronx, United States; [4]The Fleischer Institute of Diabetes and Metabolism, Albert Einstein College of Medicine, Bronx, United States; [5]Departments of Molecular Pharmacology, Albert Einstein College of Medicine, Bronx, United States

*For correspondence: jeffrey.pessin@einsteinmed.edu

Competing interest: The authors declare that no competing interests exist.

**Abstract** Cholinergic and sympathetic counter-regulatory networks control numerous physiological functions, including learning/memory/cognition, stress responsiveness, blood pressure, heart rate, and energy balance. As neurons primarily utilize glucose as their primary metabolic energy source, we generated mice with increased glycolysis in cholinergic neurons by specific deletion of the fructose-2,6-phosphatase protein TIGAR. Steady-state and stable isotope flux analyses demonstrated increased rates of glycolysis, acetyl-CoA production, acetylcholine levels, and density of neuromuscular synaptic junction clusters with enhanced acetylcholine release. The increase in cholinergic signaling reduced blood pressure and heart rate with a remarkable resistance to cold-induced hypothermia. These data directly demonstrate that increased cholinergic signaling through the modulation of glycolysis has several metabolic benefits particularly to increase energy expenditure and heat production upon cold exposure.

## Editor's evaluation

The authors report results from timely studies, demonstrating the role of TIGAR in regulating thermoregulation. Deletion of TIGAR leads to resistance to cold-induced hypothermia. The results will be of wide interest.

## Introduction

TIGAR (Tp53-induced glycolysis and apoptosis regulator) was originally identified as a p53-inducible protein that functions as a fructose-2,6-bisphosphatase (F2,6P) but subsequently shown to have phosphatase activities for a variety of phosphorylated metabolic intermediates and allosteric regulators including 2,3-bisphospholgycerate (2,3BPG), 2-phosphoglycerate, phosphoglycolate, and phosphoenolpyruvate (*Bensaad et al., 2006*; *Rigden, 2008*; *Bolaños, 2014*; *Tang et al., 2021*). Due to its F2,6P dephosphorylation activity, TIGAR is generally considered a suppressor of glycolysis as F2,6P is

a potent allosteric activator for 6-phosphofructo-1-kinase (PFK1), the rate-limiting step in glycolysis. However, the dephosphorylation of 2,3BPG to generate 3-phosphoglycerate would also be expected to increase glycolysis (*Bolaños, 2014*). In addition, the role of TIGAR in regulating carbohydrate metabolism is also complicated by the presence of the related phosphofructokinase bis-phosphatase (PFKBP) family that has both F6P 2-kinase and F2,6P bisphosphatase activities (*Mor et al., 2011*). Thus, the biological readout of TIGAR function is likely to be highly cell context dependent. In this regard, TIGAR has been reported to differentially modulate multiple different pathophysiological outcomes. For example, in the brain TIGAR expression was reported to protect against ischemic/reperfusion injury (*Li et al., 2021*), whereas in the heart TIGAR deficiency protected myocardial infarction (*Hoshino et al., 2012*) and in pressure-overloaded hearts (*Okawa et al., 2019*). In cancer models, TIGAR plays a complex role in the formation and progression of different types of cancer via suppressing aerobic glycolysis and controlling reactive oxygen species (ROS) production (*Tang et al., 2021*). Recently, it has been reported that TIGAR can both enhance the development of premalignances and suppress the metastasis of cancer invasion by the way of inhibition of ROS production (*Cheung et al., 2020*).

As carbohydrate metabolism and glycolysis are essential normal physiological processes in all cells and tissues, we have examined the metabolic phenotype of whole-body and tissue-specific TIGAR knockout mice. Surprisingly, we have observed that TIGAR deficiency in cholinergic neurons of mice results in a marked protection against hypothermia following an acute cold challenge. At a molecular level, this results from enhanced acetylcholine levels and increased cholinergic signaling at the neuro-muscular junction (NMJ) driving skeletal muscle shivering-induced thermogenesis.

## Results

Previously we reported that the TIGAR can modulate NF-kB signaling through a direct binding interaction and inhibition of the E3 ligase activity of the linear ubiquitin-binding assembly complex, LUBAC, in cultured cells (*Tang et al., 2018*). To examine the role of TIGAR in vivo, we initially generated whole-body *Tigar* knockout (TKO) mice. Surprisingly, however, during our phenotypic characterization we observed that the TKO male and female mice both display a remarkable resistance to cold induced-hypothermia. As shown in *Figure 1*, both male (panel A) and female (panel B) control mice when shifted from room temperature to a 4°C environment display the typical 4–5°C decline in core body temperature during the 1 hr acute exposure period. In contrast, the TKO male and female mice only display less than a 1°C decline in core body temperature under identical conditions. The level of *Ucp1* mRNA (*Figure 1—figure supplement 1A*) and protein (*Figure 1—figure supplement 1B*) was not significantly different in interscapular brown adipose tissue (iBAT) at room temperature. *Ucp1* mRNA did increase approximately 1.5-fold following 1 hr cold exposure, but this was not significantly different between the control and TKO mice. Although a very small amount of *Ucp1* mRNA was detected in inguinal adipose tissue (iWAT) of control mice at room temperature, it was lower in the TKO mice (*Figure 1—figure supplement 1C*). Moreover, following 4°C exposure for 2 hr there was an increase in *Ucp1* mRNA indicative of beige adipocyte induction. However, the induction of *Ucp1* mRNA was significantly reduced in the TKO mice compared to the control mice, probably due to decreased sensitivity of cold-induced sympathetic activation (see Figure 7).

Changes in tissue distribution of muscle and fat mass as well as modulation of basal energy expenditure have important roles in the control of thermogenesis. However, there was no significant difference in the relative fat mass or lean mass (*Figure 1—figure supplement 1D*) between the control and TKO mice. Hematoxylin and eosin (H&E) staining of oxidative soleus and glycolytic extensor digitorum longus (EDL) (*Figure 1—figure supplement 1E*) skeletal muscle also indicated the absence of any gross morphological differences between the control and TKO mice. Similarly, food intake (*Figure 1—figure supplement 1F*), spontaneous locomotor activity in the AMB + Z dimensions (*Figure 1—figure supplement 1G*), and energy expenditure (*Figure 1—figure supplement 1H*) were essentially identical.

As the TKO mice are whole-body knockouts, the temperature phenotype could result from an effect on a single or combinatorial number of cell types. We first generated *Tigar^{fl/fl}* mice that were crossed with *Adipoq^{Cre}* mice to generate adipocyte-specific TKO mice (*Figure 1C*). These mice had the identical temperature sensitivity as the control *Tigar^{fl/fl}* mice both displaying an approximately 4°C decline in core body temperature following exposure to 4°C for 1 hr (*Figure 1D*). To further rule out BAT as a contributor to the enhanced cold protection in the TKO mice, we crossed the cold-sensitive

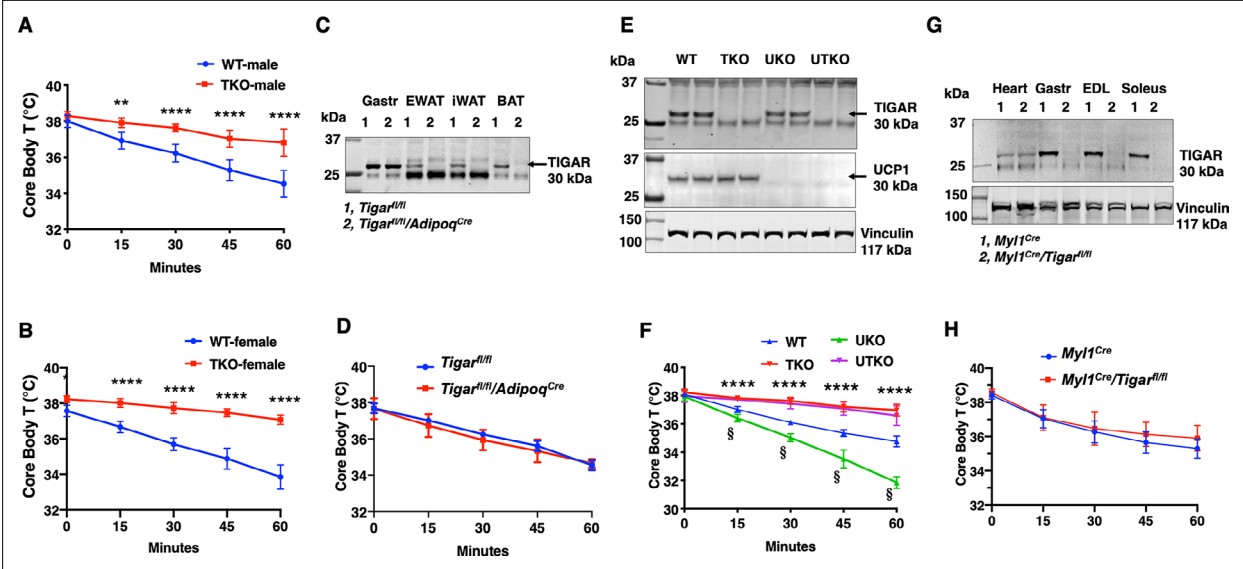

**Figure 1.** *TIGAR* deficiency mice are resistant to hypothermia induced by an acute cold challenge. (**A**) Male mice (wild type [WT] n = 10, whole-body *Tigar* knockout [TKO] n = 6) and (**B**) female mice (WT n = 6, TKO n = 7) core body temperatures were measured every 15 min starting at ambient laboratory temperature (0 min) and following placement at 4°C. (**C**) Representative TIGAR immunoblot of gastrocnemius muscle (Gastroc), epididymal white adipose tissue (EWAT), subcutaneous inguinal white adipose tissue (iWAT), and interscapular brown adipose tissue (iBAT) from the control *Tigar^fl/fl* and adipocyte-specific knockout *Tigar^fl/fl/Adipoq^Cre* mice. (**D**) *Tigar^fl/fl* and *Tigar^fl/fl/Adipoq^Cre* male mice (n = 6) core body temperatures were measured every 15 min starting at ambient laboratory temperature (0 min) and following placement at 4°C. (**E**) Representative UCP1 and TIGAR immunoblots of brown adipose tissue from WT, TKO, *Ucp1* knockout (UKO), and *Ucp1* and *Tigar* double knockout (UTKO) mice from two independent genotypes each. (**F**) WT (n = 6), TKO (n = 6), UKO (n = 7), and UTKO (n = 8) mice core body temperatures were measured every 15 min starting at ambient laboratory temperature (0 min) and following placement at 4°C. (**G**) Representative TIGAR immunoblot of heart, gastrocnemius muscle (Gastroc), extensor digitorum digitorum longus (EDL), and soleus muscle from the control (*Myl1^Cre*) and skeletal muscle-specific TKO (*Myl1^Cre/Tigar^fl/fl*) mice. (**H**) *Myl1^Cre* and *Myl1^Cr>/Tigar^fl/fl* male mice (n = 6) core body temperatures were measured every 15 min starting at ambient laboratory temperature (0 min) and following placement at 4°C. Statistical analyses are described in 'Methods details,' and the data are presented as the mean ± SD. *p<0.05, **p<0.01, ***p<0.001, ****p<0.0001, §p<0.001.

The online version of this article includes the following source data and figure supplement(s) for figure 1:

**Source data 1.** Fresh gastrocnemius muscle (Gastr), epididymal adipose tissue (EWAT), inguinal white adipose tissue, and interscapular brown adipose tissue from *Tigar^fl/fl* and *Tigar^fl/fl>/Adipoq^Cre* mice were collected, and 30 µg of the tissue lysate were used for TIGAR (30 kDa) immunoblotting analysis as described in the 'Immunoblotting' section.

**Source data 2.** Interscapular brown adipose tissues were collected from wild type (WT), whole-body *Tigar* knockout (TKO), *Ucp1* knockout (UKO), and *Ucp1* and *Tigar* double knockout (UTKO) mice, and 30 µg of the tissue lysate were used for TIGAR (30 kDa) immunoblotting analysis as described in the 'Immunoblotting' section.

**Source data 3.** Interscapular brown adipose tissues were collected from wild type (WT), whole-body *Tigar* knockout (TKO), *Ucp1* knockout (UKO), and *Ucp1* and *Tigar* double knockout (UTKO) mice, and 30 µg of the tissue lysate were used for UCP1 (33 kDa) immunoblotting analysis as described in the 'Immunoblotting' section.

**Source data 4.** Interscapular brown adipose tissues were collected from wild type (WT), whole-body *Tigar* knockout (TKO), *Ucp1* knockout (UKO), and *Ucp1* and *Tigar* double knockout (UTKO) mice, and 30 µg of the tissue lysate were used for vinculin (117 kDa) immunoblotting analysis as described in the 'Immunoblotting' section.

**Source data 5.** Fresh heart, gastrocnemius muscle (Gastr), extensor digitorum longus (EDL muscle), and soleus muscle were collected from *Myl1^Cre* and *Myl1^Cre/Tigar^fl/fl* mice, and 30 µg of the tissue lysate were used for TIGAR (30 kDa) immunoblotting analysis as described in the 'Immunoblotting' section.

**Source data 6.** Fresh heart, gastrocnemius muscle (Gastr), extensor digitorum longus (EDL muscle), and soleus muscle were collected from *Myl1^Cre* and *Myl1^Cre/Tigar^fl/fl* mice, and 30 µg of the tissue lysate were used for vinculin (117 kDa) immunoblotting analysis as described in the 'Immunoblotting' section.

**Figure supplement 1.** The whole-body *Tigar* knockout (TKO) protection against hypothermia is independent of UCP1 expression with no change in basal metabolic rate.

**Figure supplement 1—source data 1.** Interscapular brown adipose tissues were collected from both ambient temperature housed and 4°C 1 hrexposed wild type (WT) and whole-body *Tigar* knockout (TKO), mice and 30 µg of the tissue lysate were used for TIGAR (30 kDa) immunoblotting analysis as described in the 'Immunoblotting section.

*Figure 1 continued on next page*

Figure 1 continued

**Figure supplement 1—source data 2.** Interscapular brown adipose tissues were collected from both ambient temperature housed and 4°C 1 hr exposed wild type (WT) and whole-body *Tigar* knockout (TKO) mice, and 30 µg of the tissue lysate were used for UCP1 (33 kDa) immunoblotting analysis as described in the 'Immunoblotting' section.

**Figure supplement 1—source data 3.** Interscapular brown adipose tissues were collected from both ambient temperature housed and 4°C 1 hr exposed wild type (WT) and whole-body *Tigar* knockout (TKO) mice, and 30 µg of the tissue lysate were used for actin β immunoblotting analysis as described in the 'Immunoblotting' section.

**Figure supplement 2.** Changes in the expression of skeletal muscle SERCA, myoregulin, and sarcolipin do not account for the differential sensitivity to tubocurare.

*Ucp1*-deficient mice (*Enerbäck et al., 1997*) with the TKO mice (both strains with C57BL/6J background) to produce wild type (WT), TKO, UKO, and *Ucp1$^{-/-}$/Tigar$^{-/-}$* double knockout (UTKO) mice (*Figure 1E*). As expected, UKO mice were highly cold sensitive compared to their WT littermates following a 1 hr acute cold exposure (*Figure 1F*). Interestingly, the UTKO mice displayed a strong cold resistance, which was not significantly different from the TKO littermates. These data strongly suggest that thermogenic adipose tissue is not responsible for the resistance to hypothermia in the TKO mice.

Since skeletal muscle plays a major role in heat production, we next generated skeletal muscle-specific TKO mice by crossing the *Tigar$^{fl/fl}$* mice with the skeletal muscle-specific myosin light polypeptide 1-Cre (*Myl1$^{Cre}$*) mice (*Figure 1G*). TIGAR is highly expressed in skeletal muscle, with the highest protein levels in depots containing glycolytic fibers (white gastrocnemius and EDL) skeletal muscle. The *Myl1$^{Cre}$*-driven TIGAR deletion resulted in efficient loss of TIGAR protein in all skeletal muscles examined with no effect on the heart (*Figure 1G*). Similar to the adipocyte-specific TKO mice, skeletal muscle *Tigar*-deficient mice had no significant resistance to acute cold exposure (*Figure 1H*). Moreover, we did not find any significant change in skeletal muscle mRNA expression of sarco-/endoplasmic reticulum Ca$^{2+}$-ATPase (SERCA1, *Atp2a1*; SERCA2, *Atp2a2*) in control or TKO mice maintained at room temperature or shifted to 4°C for 1 hr (*Figure 1—figure supplement 2A and B*). Similarly, there was no change in myoregulin (*Mrln*) (*Figure 1—figure supplement 2C*). Although there was a small apparent increase in sarcolipin (*Sln*) mRNA in the control WT mice shifted to 4°C, there was no statistical difference with the TKO mice (*Figure 1—figure supplement 2D*). Together, these data are consistent with the absence of a cell-autonomous effect in either thermogenic adipocytes or skeletal muscle.

To unravel these unexpected findings, we next undertook a pharmacological approach to address the ability of the TKO mice to resist hypothermia. Treatment of control and TKO mice with the selective SERCA inhibitor cyclopiazonic acid (CPA) (*Wang et al., 2021*) reduced core body temperature at room temperature (*Figure 2A*). As expected, CPA treatment resulted in a faster rate of decline and to a lower extent when the mice were exposed to 4°C (*Figure 2B*). However, there was no significant difference in the response to CPA between the control and TKO mice. The CPA-induced reduction in body temperature is directly correlated with observable skeletal muscle paralysis, consistent with skeletal muscle contraction activity as an important component in thermal regulation (*Blondin and Haman, 2018*), and further indicates that SERCA functions downstream of TIGAR function.

Since *Tigar* deficiency in neither adipocytes nor skeletal muscle recapitulated the protection of acute hypothermia but that paralysis results in a rapid reduction in core body temperature, we next assessed whether the cold resistance of the TKO mice was due to shivering thermogenesis (ST). To address this, we next treated mice with the highly specific skeletal muscle nicotinic acetylcholine receptor competitive antagonist tubocurare (*Bowman, 2006*). Control male mice treated with 0.4 mg/kg tubocurare became relatively inactive with essentially no detectable locomotor activity at room temperature 10 min post injection that lasted for another 20–30 min (see *Figure 2—video 1*). Over this time period, there was a concomitant decline in core body temperature that began to recover as the effects of tubocurare began to wear off (*Figure 2C*). Tubocurare treatment of the TKO male mice at room temperature also displayed a similar decline and recovery of body temperature but that remained somewhat higher than the control mice (*Figure 2C*). However, shifting the male mice to 4°C following tubocurare injection resulted in a rapid decline in body temperature such that by 30 min the control mice's body temperature was reduced to less than 25°C (*Figure 2D*). In contrast, the tubocurare-injected TKO male mice only display a reduction in body temperature to approximately 34°C under identical conditions. Moreover, while the tubocurare-treated control mice placed at 4°C

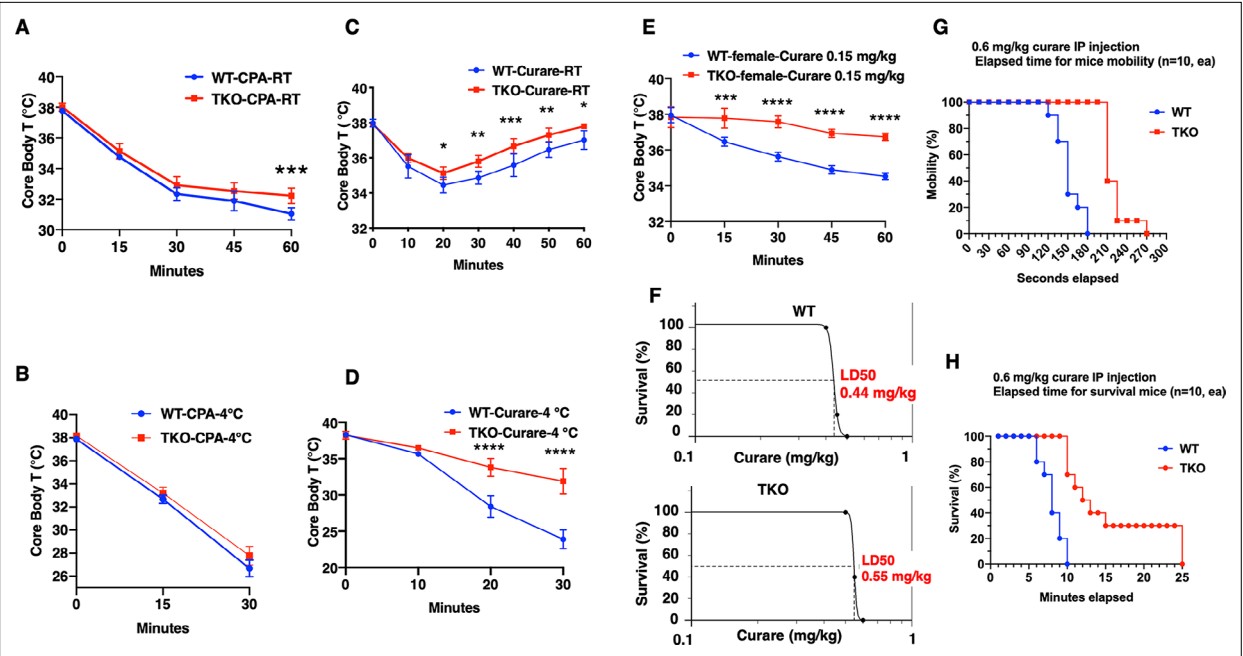

**Figure 2.** TIGAR knockout mice are resistant to tubocurare but not cyclopiazonic acid (CPA). (**A**) Wild type (WT) and whole-body *Tigar* knockout (TKO) male mice (n = 6) were intraperitoneally injected with CPA (10 mg/kg body weight), and then core body temperature was measured every 15 min at room temperature. (**B**) WT and TKO male mice (n = 6) were intraperitoneally injected with CPA (10 mg/kg body weight) at room temperature, and then shifted to 4°C for 30 min, with core body temperature measured every 15 min. (**C**) WT and TKO male mice (n = 6) were intraperitoneally injected with tubocurare (0.4 mg/kg body weight), and then core body temperature was measured every 10 minutes at room temperature. (**D**) WT and TKO male mice (n = 6) were intraperitoneally injected with tubocurare (0.4 mg/kg body weight) at room temperature, and then 10 min later shifted to 4°C for 30 min, with core body temperature measured every 10 min. (**E**) WT and TKO female mice (n = 6) were intraperitoneally injected with tubocurare (0.15 mg/kg body weight), and then core body temperature was measured every 15 min at room temperature. (**F**) WT and TKO male mice (n = 10) were intraperitoneally injected with different tubocurare doses, and the $LD_{50}$ of curare was calculated using an online software $LD_{50}$ Calculator (AAT Bioquest, Inc, Sunnyvale, CA). (**G**) WT and TKO male mice (n = 10) were intraperitoneally injected with a lethal tubocurare dose (0.6 mg/kg body weight), and the number of mice undergoing complete paralysis was plotted as a function of time in seconds. (**H**) The time to death (absence of respiration) of the same mice was plotted as a function of time in minutes. Statistical analyses are described in 'Method details,' and the data are presented as the mean ± SD. *p<0.05, **p<0.01, ***p<0.001, ****p<0.0001.

The online version of this article includes the following video and figure supplement(s) for figure 2:

**Figure supplement 1.** Changes in the expression of skeletal muscle nicotinic acetylcholine receptor subunits do not account for the differential sensitivity to tubocurare.

**Figure 2—video 1.** Whole-body *Tigar* knockout (TKO) mice are resistant to the paralytic effects of tubocurare.

https://elifesciences.org/articles/73360/figures#fig2video1

remained completely immobile, the TKO mice display apparent normal locomotor activity (***Figure 2—video 1***). Female mice are more sensitive to tubocurare than male mice (***Maurya et al., 2013***). The body temperature in WT female mice kept declining in 1 hr period after a lower dose (0.15 mg/kg) tubocurare injection at ambient temperature. Nevertheless, the decline of body temperature in the TKO female mice was resistant to the effects of tubocurare (***Figure 2E***).

The tubocurare dose-response for male mice is shown in ***Figure 2F***, with the control WT mice having an $LD_{50}$ of 0.44 mg/kg whereas the TKO mice having an $LD_{50}$ of 0.55 mg/kg (***Figure 2F***). Injection of a lethal tubocurare dose (0.60 mg/kg) initially resulted in 50% if the control mice display a loss of voluntary locomotor activity by 150 s, but this required 240 s in the TKO mice (***Figure 2G***). At 10 min, 100% of the control mice were dead, whereas it took nearly 25 min for 100% of the TKO mice to die (***Figure 2H***). The tubocurare resistance of the TKO mice was not due to the changes in expression of the nicotinic acetylcholine receptor subunits with no significant differences in the skeletal muscle expression of the α1 (*Chrna1*), β1 (*Chrnb1*), or γ (*Chrng*) subunits (***Figure 2—figure supplement 1A–C***) and with a small reduction in the d (*Chrnd*) and ε (*Chrne*) subunits (***Figure 2—figure***

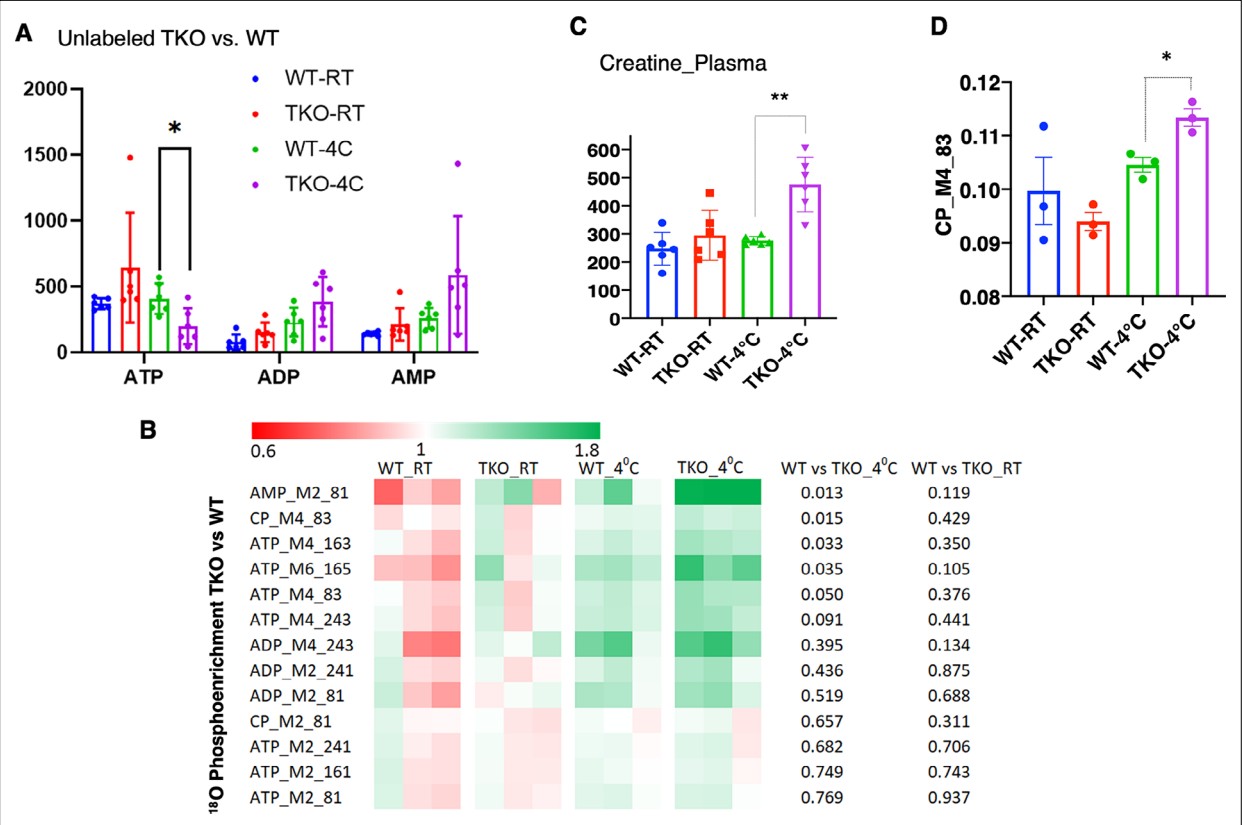

**Figure 3.** TIGAR-deficient mice display increased ATP turnover in skeletal muscle. (**A**) Wild type (WT) and whole-body *Tigar* knockout (TKO) male mice (n = 6) were maintained at room temperature (RT) or shifted to 4°C for 1 hr. Quadricep white skeletal muscles were isolated, and extracts assayed for ATP, ADP, and AMP levels as described in 'Method details.' (**B**) WT and TKO male mice (n = 3–4 per condition) were kept at RT or shifted to 4°C for 1 hr and then given an oral gavage of 0.3 ml pure $H_2O^{18}$ for 10 min at 4°C and an intraperitoneal (IP) injection of 1 ml pure $H_2O^{18}$ for another 10 min at 4°C. The quadricep muscles were freeze-clamped with liquid $N_2$ and extracts prepared for mass spectroscopic analyses. The heatmap shows the $O^{18}$ enrichment fraction of ATP, ADP, AMP, and creatine-phosphate. (**C**) WT and TKO male mice (n = 6) were maintained at RT or shifted to 4°C for 1 hr, and plasma creatine levels were determined by metabolomics analyses. (**D**) The CP_M4_83 data was generated from the same experimental setting as (**B**), indicating the increase in phosphocreatine turnover in TKO compared to that in WT at 4°C. Statistical analyses are described in 'Method details,' and the data are presented as the mean ± SD. *p<0.05, ***p<0.01.

The online version of this article includes the following source data and figure supplement(s) for figure 3:

**Source data 1.** Raw $^{18}O$ enrichment data of stable isotope metabolic flux assessment with $H_2^{18}O$ was collected in quadricep white muscle of both ambient temperature housed and 4°C 1 hr exposed wild type (WT) and whole-body *Tigar* knockout (TKO) mice, as described in 'Method details'.

**Figure supplement 1.** Skeletal muscles of whole-body *Tigar* knockout (TKO) mice at 4°C display increased pentose phosphate pathway, purine nucleotide cycle, and amino acid utilization pathways.

**Figure supplement 1—source data 1.** Raw metabolites data were collected in the gastrocnemius muscle of both ambient temperature housed and 4°C 1 hr exposed wild type (WT) and whole-body *Tigar* knockout (TKO) mice as described in 'Method details'.

**Figure supplement 2.** TIGAR deficiency has no significant effect on intrinsic skeletal muscle mitochondrial oxidative phosphorylation activity or ATP production.

*supplement 1D and E*) in the TKO mice. These data suggest that the protection against hypothermia was due to cholinergic signaling to the skeletal muscle.

Consistent with increased skeletal muscle energetic demand, ATP levels were significantly decreased with an apparent increase in ADP and AMP in the TKO mice after 1 hr at 4°C (*Figure 3A*). Stable isotope metabolic flux assessment with $H_2O^{18}$ demonstrated increased incorporation of ortho-phosphate into ATP, ADP, and AMP in the TKO mice at 4°C, which was not significantly different at room temperature (*Figure 3B*). Plasma creatine was also increased twofold in the TKO at 4°C (*Figure 3C*), reflecting the increased phosphocreatine turnover (CP M4 phosphate fragment *m/z* 83, *Figure 3D*). The appearance of creatine in the plasma is likely due, in part, to the increased rate of shivering that is known to induce skeletal muscle leakage (*Baird et al., 2012*).

The mechanics of energetic supply at 4°C was further examined by using a widely targeted LC/MS/MS protocol (*Hoshino et al., 2012*). Key pathways found to be significant were pentose phosphate pathway, purine nucleotide cycle, and amino acid utilization pathways (*Figure 3—figure supplement 1A*). The pentose phosphate pathway cycle fills the purine nucleotide cycle that acts in concert with the malate-aspartate shuttle and the TCA cycle to preserve cellular energetics (*Hatazawa et al., 2015*). In particular, glycine/serine, histidine, and methionine metabolism also drives the input of the tricarboxylic acid cycle intermediates serving as anapleurotic inputs (*Figure 3—figure supplement 1B*). Together, these interactions serve to recycle AMP generated by increased skeletal muscle energy demand.

Mitochondria are a primary source of both ATP production and heat generation during muscle contraction. To determine whether mitochondrial oxidative phosphorylation (OXPHOS) efficiency and/or capacity are altered in TKO mice, permeabilized skeletal muscle fiber bundles were prepared from red portions of the gastrocnemius muscle from the control and TKO mice immediately after 1 hr cold exposure. ADP-stimulated respiratory kinetics (i.e., $K_m$ and $V_{max}$) were virtually identical between WT and TKO muscle during respiration supported by carbohydrate (*Figure 3—figure supplement 2A and B*) and/or lipid-based substrates. Rates of ATP production (JATP) and oxygen consumption (JO₂) measured simultaneously under clamped submaximal and maximal ADP-demand states were also similar in muscle from the control and TKO mice (*Figure 3—figure supplement 2C and D*), yielding similar ATP/O efficiency ratios (*Figure 3—figure supplement 2E*). Finally, in vitro studies of the EDL muscle showed no differences in force-frequency, peak specific tension, or time to one-half relaxation between the control and TKO mice (*Figure 3—figure supplement 2F and G*). Together, these data indicate that the skeletal muscle contraction-induced thermogenesis in TKO mice is not due to any intrinsic change in skeletal muscle mitochondrial efficiency of contractile function and therefore must result from a change in signaling that in turn drives an enhanced skeletal muscle response.

Since the TKO mice are resistant to tubocurare and the major driver of skeletal muscle contraction is cholinergic signaling, we next generated cholinergic neuron-specific *Tigar* knockout (chTKO) mice using the choline acetyl-CoA transferase *ChAT-Cre* mice (*Rossi et al., 2011*). As the superior cervical ganglion (SCG) is composed of approximately 80% cholinergic neurons (*Wang et al., 1990*; *Morales et al., 1995*; *Juranek and Wojtkiewicz, 2015*), we collected the fresh SCGs (*Figure 4—figure supplement 1*) and confirmed that *Chat^Cre* resulted in the loss of *Tigar* in cholinergic neurons by immunofluorescence co-localization of TIGAR with the vesicular choline transporter (VChAT) (*Figure 4A*) and by immunoblotting analyses of TIGAR protein in the SCGs (*Figure 4B*). Similar to the TKO mice, the chTKO mice were also resistant to acute cold exposure compared to the control *Chat^Cre* mice (*Figure 4C*). Treatment with the SERCA inhibitor CPA, at room temperature (*Figure 4D*) or at 4°C (*Figure 4E*), also resulted in paralysis and a decline in core body temperature that was not significantly different between the *Chat^Cre* and chTKO mice, although the reduction in core body temperature was much greater at 4°C than at room temperature. The chTKO mice were also resistant to tubocurare at room temperature with both less of a core body temperature drop and a faster recovery (*Figure 4F*). In agreement with the TKO mice (*Figure 2D*), the resistance to cold-induced hypothermia by acute 4°C challenge was very dramatic in the chTKO mice compared to the control mice (*Figure 4G*). Similarly, the chTKO mice were highly resistant to the paralytic actions of tubocurare (*Figure 4—video 1*). Moreover, immunoblotting of the SCG for the neuron activation marker c-Fos further demonstrated that the control mice displayed a small increase in c-Fos protein levels when cold challenged that was further increased in the chTKO mice (*Figure 4H*). As controls, there were no significant differences in the protein levels of the vesicular acetylcholine transporter (VAChT) or the plasma membrane choline transporter (ChT) proteins. These data demonstrate that the cholinergic neuron TKO mice recapitulate the cold resistance and pharmacological characteristics of the whole-body TKO mice, consistent with these mice displaying increased cholinergic tone.

If increased cholinergic input is responsible for the increased energy demand and heat production by skeletal muscle, then the skeletal muscle metabolic characteristics of the TKO mice should be recapitulated in the chTKO mice. As shown in *Figure 5*, widely targeted LC/MS/MS in the gastrocnemius muscle of the chTKO mice identified the same three key pathways – pentose phosphate pathway, purine nucleotide cycle, and amino acid utilization pathways – that display a greater induction at 4°C. Together, these data are consistent with increased cholinergic signaling to skeletal muscle that is responsible for the increased skeletal muscle thermogenesis following cold exposure.

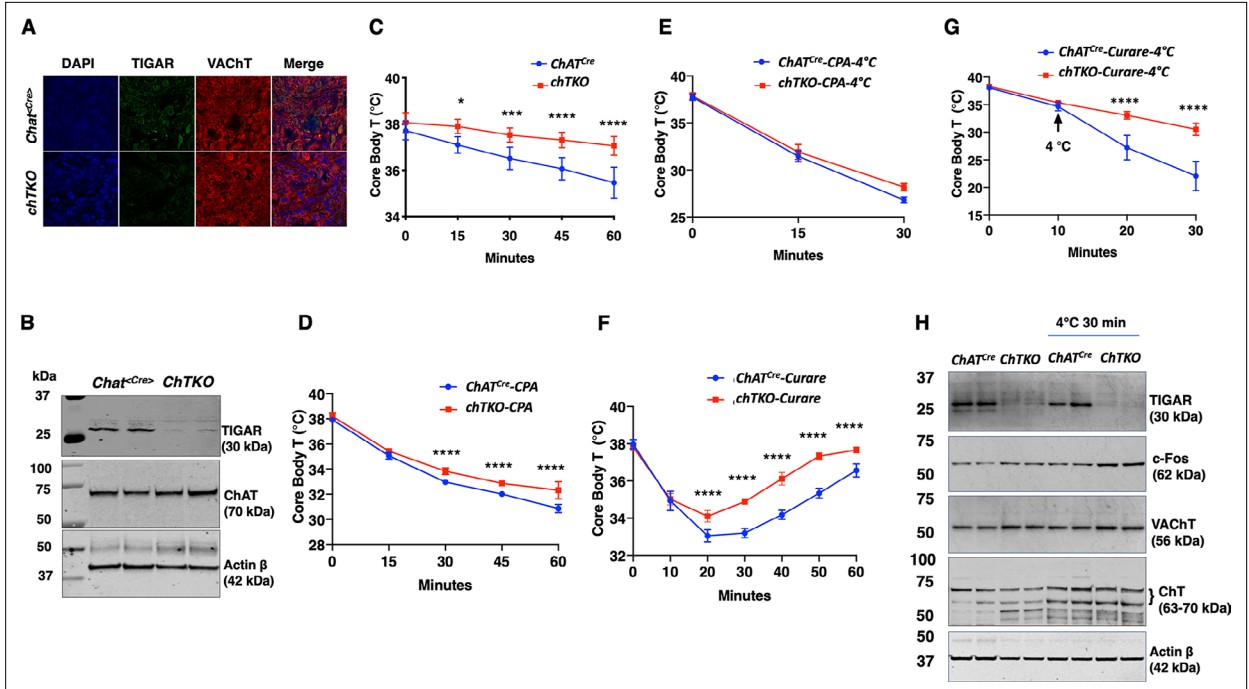

**Figure 4.** Cholinergic neuron-specific TIGAR knockout mice recapitulate the protection against hypothermia of the whole-body TIGAR-deficient mice. (**A**) Representative immunofluorescent images showing TIGAR, vesicular acetylcholine transporter (VAChT), and nuclei (DAPI) in the superior cervical ganglions (SCGs) from the control *Chat^Cre* and cholinergic neuron-specific *Tigar* knockout (chTKO) male mice. (**B**) Representative TIGAR, choline acetyltransferase (ChAT), and actin protein immunoblots of the SCG from two independent control *Chat^Cre* and chTKO male mice. (**C**) Male *Chat^Cre* and chTKO mice (n = 6) core body temperatures were measured every 15 min starting at ambient laboratory temperature (0 min) and following placement at 4°C. (**D**) Male *Chat^Cre* and chTKO mice (n = 6) were intraperitoneally injected with cyclopiazonic acid (CPA,10 mg/kg body weight) at room temperature and core body temperature measured every 15 min. (**E**) *Chat^Cre* and chTKO male mice (n = 6) were intraperitoneally injected with CPA (10 mg/kg body weight) at room temperature, shifted to 4°C, and core body temperature measured every 15 min. (**F**) Male *Chat^Cre* and chTKO mice (n = 6) were intraperitoneally injected with tubocurare (0.4 mg/kg body weight), and then core body temperature was measured every 15 min at room temperature. (**G**) *Chat^Cre* and chTKO mice (n = 6) were intraperitoneally injected with tubocurare (0.4 mg/kg body weight) at room temperature and 10 min later shifted to 4°C for 30 min. Core body temperature was measured every 10 minutes. (**H**) Representative TIGAR, c-Fos, VAChT, ChT, and actin protein immunoblots of the SCG from two independent control *Chat^Cre* and chTKO male mice at room temperature or shifted to 4°C for 30 min. Statistical analyses are described in 'Method details,' and the data are presented as the mean ± SD. *p<0.05, **p<0.01, ***p<0.001, ****p<0.0001.

The online version of this article includes the following video, source data, and figure supplement(s) for figure 4:

**Source data 1.** The superior cervical ganglion (SCG) tissues were collected and snap-frozen in liquid nitrogen from the *Chat^Cre* and cholinergic neuron-specific *Tigar* knockout (chTKO) mice, as shown in *Figure 4—figure supplement 1* for SCG dissection.

**Source data 2.** The superior cervical ganglion (SCG) tissues were collected and snap-frozen in liquid nitrogen from the *Chat^Cre* and cholinergic neuron-specific *Tigar* knockout (chTKO) mice, as shown in *Figure 4—figure supplement 1* for SCG dissection.

**Source data 3.** The superior cervical ganglion (SCG) tissues were collected and snap-frozen in liquid nitrogen from the *Chat^Cre* and cholinergic neuron-specific *Tigar* knockout (chTKO) mice, as shown in *Figure 4—figure supplement 1* for SCG dissection.

**Source data 4.** The superior cervical ganglion (SCG) tissues were collected and snap-frozen in liquid nitrogen from both ambient temperature housed and 4°C 30 min exposed *Chat^Cre* and cholinergic neuron-specific *Tigar* knockout (chTKO) mice, as shown in *Figure 4—figure supplement 1* for SCG dissection.

**Source data 5.** The superior cervical ganglion (SCG) tissues were collected and snap-frozen in liquid nitrogen from both ambient temperature housed and 4°C 30 min exposed *Chat^Cre* and cholinergic neuron-specific *Tigar* knockout (chTKO) mice, as shown in *Figure 4—figure supplement 1* for SCG dissection.

**Source data 6.** The superior cervical ganglion (SCG) tissues were collected and snap-frozen in liquid nitrogen from both ambient temperature housed and 4°C 30 min exposed *Chat^Cre* and cholinergic neuron-specific *Tigar* knockout (chTKO) mice, as shown in *Figure 4—figure supplement 1* for SCG dissection.

**Source data 7.** The superior cervical ganglion (SCG) tissues were collected and snap-frozen in liquid nitrogen from both ambient temperature housed and 4°C 30 min exposed *Chat^Cre* and cholinergic neuron-specific *Tigar* knockout (chTKO) mice, as shown in *Figure 4—figure supplement 1* for SCG dissection.

*Figure 4 continued on next page*

*Figure 4 continued*

**Source data 8.** The superior cervical ganglion (SCG) tissues were collected and snap-frozen in liquid nitrogen from both ambient temperature housed and 4°C 30 min exposed *Chat^Cre* and cholinergic neuron-specific *Tigar* knockout (chTKO) mice, as shown in *Figure 4—figure supplement 1* for SCG dissection.

**Figure supplement 1.** Mouse superior cervical ganglion (SCG) dissection.

**Figure 4—video 1.** Cholinergic neuron-specific *Tigar* knockout (chTKO) mice are resistant to the paralytic effects of tubocurare.
https://elifesciences.org/articles/73360/figures#fig4video1

To address the basis of the apparent increase in cholinergic signaling, we first asked whether there was a change in synaptic density in the NMJ. The acetylcholine receptor aggregates in the synaptic cleft and can be specifically labeled with α-bungarotoxin (*Pratt et al., 2018*), as shown for the whole EDL muscle (*Figure 6A*). Quantification revealed a large degree of heterogeneity in the number of clustered acetylcholine receptors per muscle (*Figure 6B*). Nevertheless, there was a clear statistical increase in the TKO mice compared to the control mice. Consistent with the increase in synaptic junctions, the total amount of acetylcholine was also increased in skeletal muscle extracts from the TKO and chTKO mice compared to their respective control mice at both room temperature and after cold exposure (*Figure 6C and D*).

As it is not possible to determine the rate of acetylcholine synthesis in vivo, we took advantage of SH-SY5Y cells that have characteristics of cholinergic neurons (*Kovalevich and Langford, 2013*). Using sgRNA and *Cas9*, we effectively knocked out *Tigar* in several clonal SH-SY5Y cells, two representative control, and *sgRNA* knockout cells as shown in *Figure 6E*. The amount of acetylcholine secreted

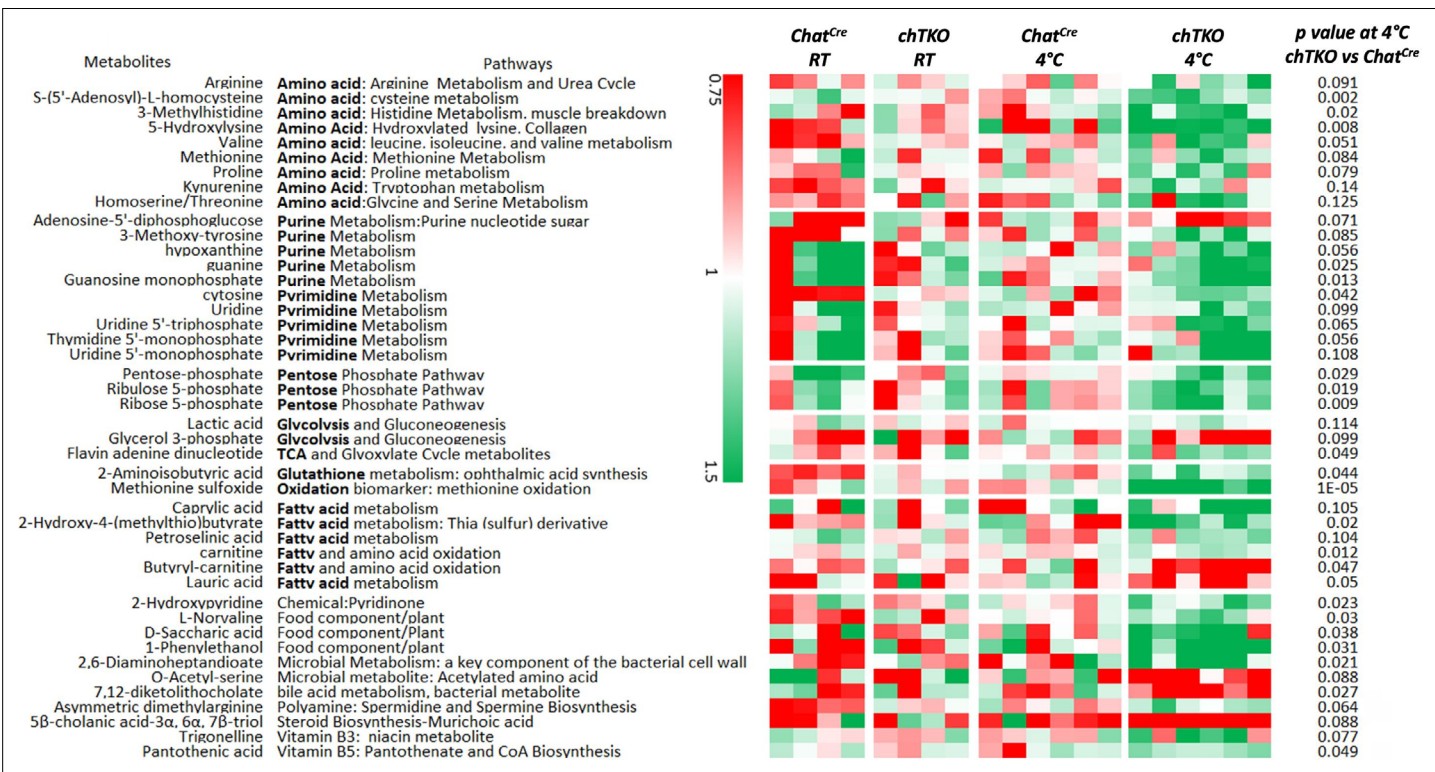

**Figure 5.** Skeletal muscles of the cholinergic neuron-specific *Tigar* knockout (chTKO) mice at 4°C display increased pentose phosphate pathway, purine nucleotide cycle, and amino acid utilization pathways. The control *Chat^Cre* and chTKO male mice (n = 4–6) were maintained at room temperature (RT) or shifted to 4°C for 1 hr. Quadricep white skeletal muscles were isolated and extracts were subjected to widely targeted (multiple reaction monitoring [MRM]) small metabolite profiling using an ABSciex 6500 + QTRAP with ACE PFP and Merck ZIC-pHILIC columns as described in 'Materials and methods.' The heatmap shows the metabolites/pathways differentially identified with the corresponding p-values.

The online version of this article includes the following source data for figure 5:

**Source data 1.** Raw metabolites data were collected in the gastrocnemius muscle of both ambient temperature housed and 4°C 1 hr exposed *Chat^Cre* and cholinergic neuron-specific *Tigar* knockout (chTKO) mice as described in 'Method details'.

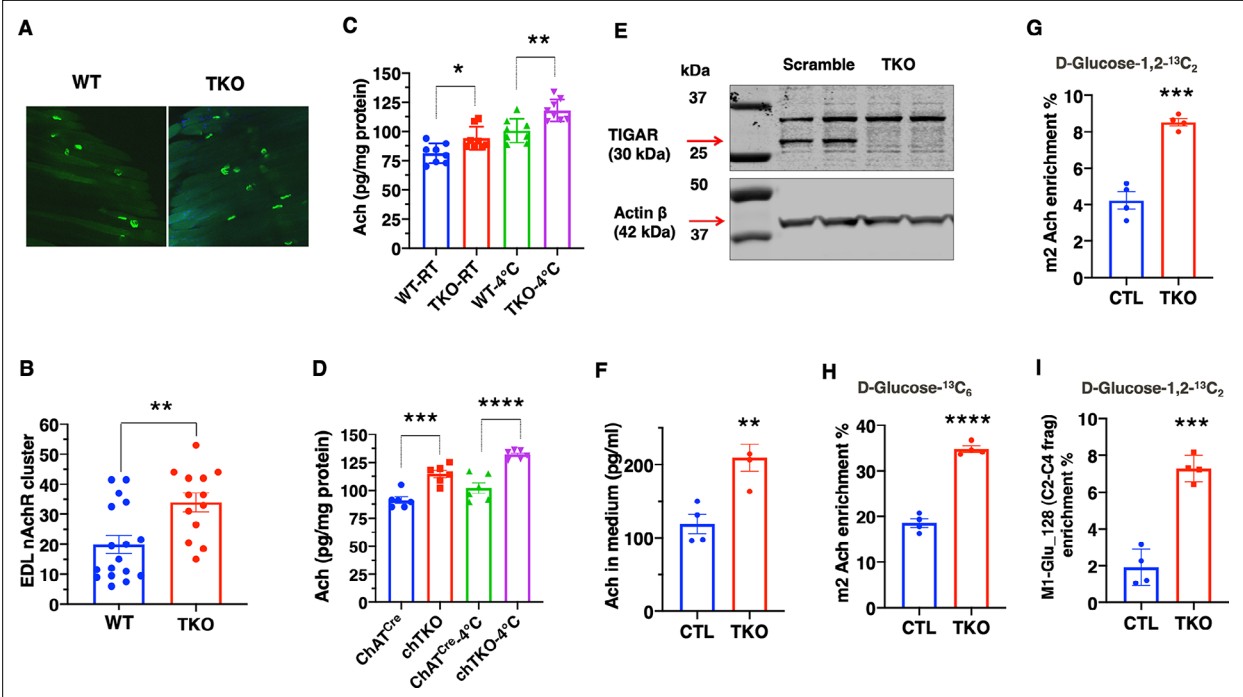

**Figure 6.** TIGAR deficiency increases acetylcholine biosynthesis and acetylcholine receptor clustering at the neuromuscular junction. (**A**) Representative images of α-bungarotoxin immunofluorescent labeling of nicotinic acetylcholine receptor clusters in the extensor digitorium longus (EDL) muscle from wild type (WT) and whole-body *Tigar* knockout (TKO) mice. (**B**) Quantification of the number of nicotinic acetylcholine receptor clusters following 15 min exposure at 4°C (WT n = 17, TKO n = 13). These data represent the average of over six mice in each group of mean ± SD (unpaired *t*-test, two-tailed, **p=0.0035). (**C**) Acetylcholine levels in the gastrocnemius muscle of WT and TKO male (n = 7) mice at room temperature or following 1 hr at 4°C. (**D**) Acetylcholine levels in the gastrocnemius muscle of *Chat^Cre* and chTKO male (n = 6) mice at room temperature or following 1 hr at 4°C. (**E**) Representative immunoblots of TIGAR and actin proteins from two scrambled sgRNA and two *Tigar* sgRNA knockout Sh-SY5Y cell lines. (**F**) Acetylcholine concentrations in the medium of scrambled and TKO SH-SH5Y neuroblastoma cells. (**G**) m2 acetylcholine enrichments in cells labeled with D-glucose-1,2-$^{13}$C$_2$ (**H**) m2 acetylcholine enrichments in cells labeled with U-$^{13}$C$_6$ D-glucose. (**I**) m1 glutamate (*m/z* 128, C2-C4 fragment) enrichment in the medium of scrambled and TKO SH-SY5Y human neuroblastoma cells labeled with D-glucose-[1,2]-$^{13}$ $_{C2}$. Statistical analyses are described in 'Method details,' and the data are presented as the mean ± SD. *p<0.05, ***p<0.001, ****p<0.0001.

The online version of this article includes the following source data and figure supplement(s) for figure 6:

**Source data 1.** The culture SH-SY5Y cells were collected from both scrambled and whole-body *Tigar* knockout (TKO) cells, and 30 µg of the cell lysates were used for TIGAR (30 kDa) immunoblotting analysis as described in the 'Immunoblotting' section.

**Source data 2.** The culture SH-SY5Y cells were collected from both scrambled and whole-body *Tigar* knockout (TKO) cells, and 30 µg of the cell lysates were used for actin β (42 kDa) immunoblotting analysis as described in the 'Immunoblotting' section.

**Figure supplement 1.** Summation of flux results through glycolysis and the TCA cycle using [1,2]–$^{13}$C-glucose by assessing glutamate isotopomers.

**Figure supplement 1—source data 1.** Raw metabolites data including cellular acetyl-CoA and acetyl-carnitine were collected from SH-SY5Y control and whole-body *Tigar* knockout (TKO) cell pellet as described in 'Method details'.

into the medium over a 24 hr period was increased in the TKO SH-SY5Y cells compared to controls (*Figure 6F*). Labeling with [1,2]-$^{13}$C glucose demonstrates an increase in the m2-labeled acetylcholine levels in the TKO SH-SY5Y cells (*Figure 6G*). Similarly, labeling with U-$^{13}$C glucose also demonstrated an increase in m2-labeled acetylcholine in the TKO SH-SY5Y cells (*Figure 6H*). Moreover, there was a threefold increase in the enrichment of the C2-C4 fragment, m1-glutamate *m/z* 129, indicating that the acetyl group in acetylcholine originated from glycolysis via pyruvate dehydrogenase (PDH) (*Figure 6I*, *Figure 6—figure supplement 1A and B*). This is because the m2 acetyl group formed via PDH generates $^{13}$C atoms at positions 4 and 5 of glutamate, whereas the m2 $^{13}$C glutamate (*m/z* 130) at positions 2 and 3 is derived from pyruvate carboxylase (PC)-generated oxaloacetate (*Madhu et al., 2020*) schematically shown in *Figure 6—figure supplement 1A*. Analyses of the *m/z* 128–130 C2-C4 glutamate fragment distinguish these isotopomers with the enrichment of m1 glutamate (129 fragment) approximately five times more abundant than the m2 glutamate (130 fragment) in the TKO cells (*Figure 6—figure supplement 1B*).

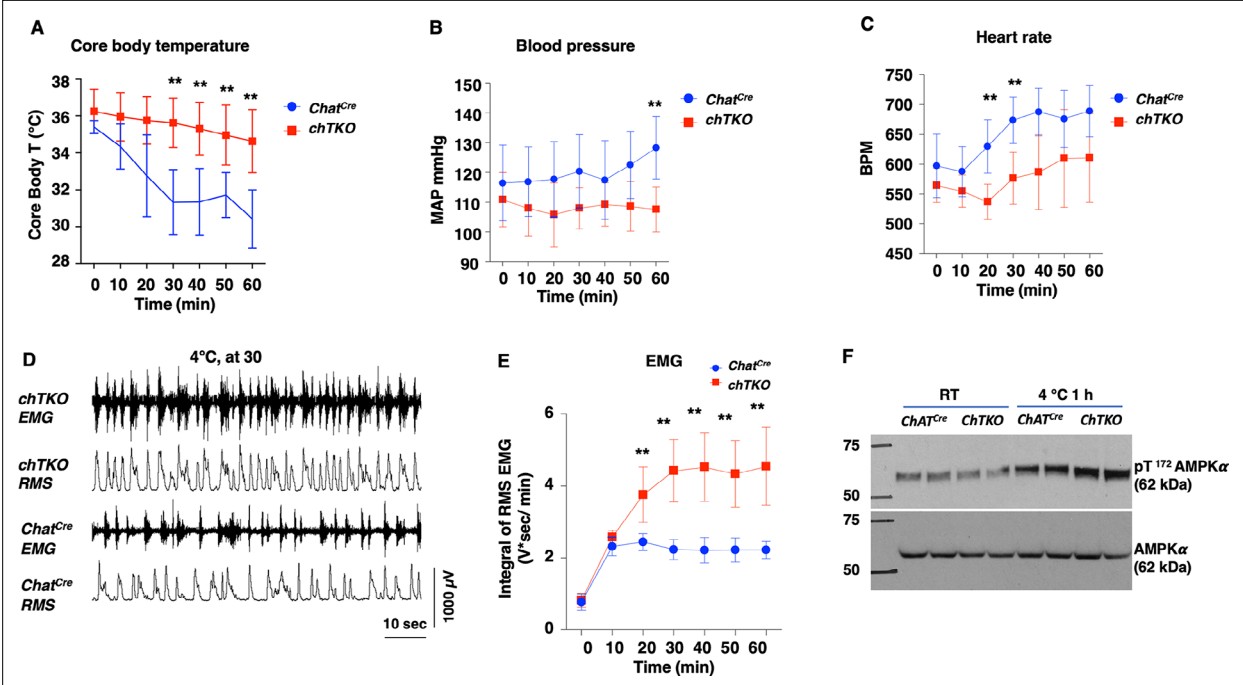

**Figure 7.** The cholinergic neuron-specific *Tigar* knockout (chTKO) mice display enhanced cold-stimulated skeletal muscle shivering activity and increased cholinergic tone. (**A**) Male mice core body temperatures (*Chat^Cre* n = 7, TKO n = 10), (**B**) blood pressure (mmHg) (*Chat^Cre* n = 7, chTKO n = 10), (**C**) heart rate (beats per minute, BPM) (*Chat^Cre* n = 7, chTKO n = 6), (**D**) representative electromyography (EMG) traces and root mean squares (RMS) for *Chat^Cre* and chTKO male mice when shifted to 4°C for 30 min, and (**E**) neck EMG (*Chat^Cre* n = 5, chTKO n = 6) were measured continuously starting at ambient laboratory temperature (0 min) and for 60 min following placement at 4°C. (**F**) Extensor digitorium longus (EDL) muscles from two independent control *Chat^Cre* and chTKO male mice at room temperature (RT) or shifted to 4°C for 1 hr were isolated, and tissue extracts were immunoblotted with an antibody for the phospho-threonine 172 AMPKα subunit (pT172-AMPKα) or total α subunit (AMPKα). Statistical analyses are described in 'Method details,' and the data are presented as the mean ± SD. **p<0.01.

The online version of this article includes the following source data for figure 7:

**Source data 1.** The extensor digitorum longus (EDL) muscles were collected from both ambient temperature housed and 4°C 1 hr exposed *Chat^Cre* and cholinergic neuron-specific *Tigar* knockout (chTKO) mice, and 30 µg of the tissue lysate were used for AMPKα (62 kDa) immunoblotting analysis as described in the 'Immunoblotting' section.

**Source data 2.** The extensor digitorum longus (EDL) muscles were collected from both ambient temperature housed and 4°C 1 hr exposed *Chat^Cre* and cholinergic neuron-specific *Tigar* knockout (chTKO) mice, and 30 µg of the tissue lysate were used for phospho-^Thr172AMPKα (62 kDa) immunoblotting analysis as described in the 'Immunoblotting' section.

If the changes in glucose flux occurred through increased flux through PDH, then we should also observe a concomitant increase in the levels of acetyl-CoA that is directly generated by PDH from pyruvate. As shown in *Figure 6—figure supplement 1C*, acetyl-CoA levels were increased approximately 1.5-fold and acetyl-carnitine levels approximately 2.5-fold (*Figure 6—figure supplement 1D*). Since acetyl-CoA is converted to acetyl-carnitine in the mitochondria, these demonstrate that *Tigar* deficiency not only increases glycolysis but also results in increased mitochondria uptake of acetyl-CoA (via PDH). The increase in mitochondria PDH flux would in turn drive an increase in oxygen consumption. Seahorse flux analyses of oxygen consumption rate (OCR) clearly demonstrated that the TKO SH-SY5Y cells had higher rates of OCR compared to the control SH-SY5Y cells (*Figure 6—figure supplement 1E*). Together, these data demonstrate that *Tigar* deficiency in SH-SY5Y neuroblastoma cells results from an increase in intracellular acetyl-CoA levels consistent with increased glycolysis and direct conversion of pyruvate to acetyl-CoA by PDH, as opposed to PC. It is the increase in PDH flux into the tricarboxylic acid cycle that drives increased mitochondria oxidation.

A generalized increased cholinergic tone would be expected to reduce blood pressure and heart rate but with increased skeletal muscle shivering. To assess these physiological parameters, we used implanted telemetry probes to simultaneously determine body temperature, mean arterial pressure (MAP), heart rate (beats per minute [BPM]), and skeletal muscle electromyography (EMG) in the

cold-exposed *Chat^Cre* and chTKO mice (*Figure 7*). Consistent with the rectal temperature measurements previously observed, the telemetry probe also demonstrated that the chTKO mice are more resistant to cold-induced hypothermia than *Chat^Cre* mice (*Figure 7A*). At room temperature, there was no significant difference in blood pressure, but following cold exposure the blood pressure of the control mice increased, whereas the blood pressure of the chTKO mice decreased such that by 60 min the chTKO mice had a blood pressure of 25 mmHg less than the control mice (*Figure 7B*). Similarly, the heart rate was not statistically different at room temperature but increased substantially more in the control mice than in the chTKO mice following cold exposure (*Figure 7C*). As these data strongly suggest increased cholinergic signaling to enhance skeletal muscle contraction, we utilized EMG to directly determine skeletal muscle activity. A representative image of EMG tracers and root mean square (RMS) derivation is shown for the *Chat^Cre* and chTKO mice when shifted to 4°C for 30 min (*Figure 7D*). Quantification of these types of data as the integral of RMS during the time course of cold exposure demonstrated that at room temperature and following 10 min of cold exposure there is no significant difference in skeletal muscle contraction activity between the *Chat^Cre* and chTKO mice (*Figure 7E*). However, the contraction activity of the control *Chat^Cre* mice plateau at this level, whereas the chTKO mice shivering activity further increases to nearly twice the level of the control mice.

The increased skeletal muscle shivering associated with increased ATP turnover and energy expenditure would be expected to result in a compensatory increase in AMP-dependent protein kinase (AMPK) activity. As shown in *Figure 7F*, there was no significant difference in the activation site phosphorylation (T172) of the AMPK α subunit in the EDL muscle at room temperature between the control *Chat^Cre* and chTKO mice. However, after 1 hr at 4°C, the control mice display the expected increase in T172-AMPKα phosphorylation that was further increased in the chTKO mice. Together, the relative changes in cold--induced blood pressure, heart rate, skeletal muscle shivering, and increase in AMPK activation are fully consistent with the chTKO mice displaying a generalized increase in cholinergic signaling following cold exposure.

## Discussion

Currently, there is a substantial effort to understand and develop approaches to increase energy expenditure through the development/activation of brown/beige adipose tissue thermogenesis. A fraction of adult humans can respond to cold stress by inducing the expression of thermogenic adipose tissue under relatively long-term cold acclimation in adult humans (*McNeill et al., 2021*). However, it remains somewhat controversial if this inducible thermogenic human brown adipose tissue is the rodent equivalent of beige or brown adipocytes (*Cannon et al., 2020*; *Samuelson and Vidal-Puig, 2020*; *Virtanen and Nuutila, 2021*). In addition, whether the mass of the human-inducible thermogenic adipocytes is sufficient to significantly contribute to cold-responsive thermogenesis for core temperature maintenance or weight loss has also been questioned (*Blondin et al., 2014*; *Virtanen and Nuutila, 2021*). In contrast, skeletal muscle accounts for approximately 40% of healthy body mass and is the dominant factor for basal metabolic rate and the major driver of energy expenditure during physical exercise (*Haman et al., 2010*; *Blondin et al., 2014*). In rodents, non-shivering thermogenesis (NST) is driven by UCP1-dependent mitochondrial uncoupling in both brown and beige adipocytes by the creatine phosphorylation/dephosphorylation cycle in beige adipocytes (*Cohen and Spiegelman, 2015*; *Kazak et al., 2015*; *Jung et al., 2019*). There are also reports of a skeletal muscle-based NST via a sarcolipin-dependent futile cycle of SERCA-dependent calcium transport in the sarcoplasmic reticulum (*Bal et al., 2012*; *Maurya et al., 2015*). In humans, maximal NST can increase energy production equivalent to about two times above the resting metabolic rate (RMR), while moderate ST is 2.5–3.5 times RMR and maximal shivering has been measured at about five times RMR (*Haman et al., 2010*; *Blondin et al., 2014*). In contrast, exercise at ~50% VO$_2$ max generates approximately 15 times RMR (*Weber and Haman, 2005*; *Blondin et al., 2014*). While a substantial effort has focused on NST, shivering and physical activity are the dominant forms of involuntary and voluntary heat production during acute cold exposure. Thus, it is essential to develop a deeper understanding of the physiological integrated role of skeletal muscle with thermogenic adipose tissue to have a complete and detailed molecular understanding of energy balance and its implications for potential interventions to promote and maintain weight loss.

The data presented in this article demonstrate that cholinergic neuron deficiency of *Tigar* results in an enhancement of cold-induced skeletal muscle contraction/shivering that is responsible for an

increase in heat generation and protection of acute cold-induced hypothermia. This response is likely mediated by an increase acetylcholine signaling at the NMJ, consistent with the parallel induction of tubocurare resistance and increased cholinergic neuron activation. Interestingly, despite the increase in synaptic density and acetylcholine levels of mice at room temperature that are mildly cold stress (thermoneutrality for mice is approximately 29–31°C; *Ganeshan and Chawla, 2017*), *Tigar* deficiency does not appear to significantly affect body temperature, skeletal muscle contraction activity, or the overall skeletal muscle metabolite profile at room temperature. However, following acute cold challenge there are marked changes in these parameters between WT and cholinergic TKO mice including a metabolic profile indicative of increased ATP turnover. These findings suggest that there is an increase in cholinergic signaling reserve that only becomes apparent when cholinergic signaling demand is high.

In addition to the well-established role of sympathetic activation to increase brown and beige adipocytes thermogenesis (*Zhu et al., 2019*), both cholinergic and sympathetic activation can affect vascular blood flow and behavorial activities that also contribute to body temperature regulation (*Nakamura et al., 2022*). Although we cannot rule out the contribution of vasoconstriction to the conservation of core body temperature under cold exposure, heat loss from the mouse tail only accounts for 5–8% of whole-body heat loss (*Skop et al., 2020*) and is therefore unlikely to play a significant role in the thermoregulation observed by cholinergic *Tigar* deficiency.

In addition to TIGAR as a small carbohydrate phosphatase, previous studies have shown that TIGAR can directly interact with hexokinase 2 and the HOIP subunit of the linear ubiquitin assembly complex (*Cheung et al., 2012*; *Tang et al., 2018*). Thus, although it is formally possible that TIGAR increases cholinergic signaling due to alterations in protein-protein interactions, we believe that it is more likely a result of increased glycolysis due to its phosphatase function that is occurring in the cholinergic neurons as this would result in an increase in acetyl-CoA levels. The steady-state acetyl-CoA concentrations in neuronal mitochondrial and cytosol are several folds lower than the acetyl-CoA Km values for choline acetyltransferase. More specifically, following cholinergic differentiation there is a further decrease in acetyl-CoA levels that results in a further substrate dependence of acetylcholine biosynthesis (*Ronowska et al., 2018*). In contrast, several studies have also shown that choline uptake via the choline transporter is rate limiting for acetylcholine synthesis (*Lockman and Allen, 2002*). Although we have not measured choline uptake activity, *Tigar* deficiency had no significant effect on the amount of choline transporter protein. However, metabolic analyses clearly demonstrated an increased glucose-derived acetyl moiety into acetylcholine and increased steady-state acetylcholine levels with increased acetyl-CoA and acetyl-carnitine levels.

Thus, the simplest and most direct mechanism for the increase in acetylcholine levels induced by *Tigar* deficiency is the increased formation of acetyl-CoA through enhanced glycolysis, which in turn leads to increased production of acetylcholine in cholinergic neurons. At present, it is not possible to distinguish the amount of synaptic acetyl-CoA from skeletal muscle acetyl-CoA at the NMJ, but future studies using spatial mass spectroscopy (MALDI) may be able to resolve this issue.

Finally, it is also important to recognize that myasthenia gravis is a neuromuscular disease caused by autoantibodies against components of the NMJ and in particular the nicotinic acetylcholine receptor (*Gilhus et al., 2019*). Current therapeutic strategies are focused on the use of steroids or immune suppressants to block the immune system and drugs to increase the transmission of acetylcholine signaling such as acetylcholine esterase inhibitors (*Gilhus et al., 2019*). Similarly, central cholinergic neurons undergo severe neurodegeneration in Alzheimer's disease and agents that enhance cholinergic signaling, particularly cholinesterase inhibitor therapies, providing significant symptomatic improvement in patients with Alzheimer's disease (*Ferreira-Vieira et al., 2016*; *Ahmed et al., 2019*). Interestingly, recent reports show that the NMJs are stable in human age-related sarcopenia and in patients with cancer cachexia, which are different from the rodent models in that the sarcopenia is linked with the endplate fragmentation in NMJ (*Jones et al., 2017*; *Boehm et al., 2020*). As such, inhibition of TIGAR function may also provide a novel approach to improve skeletal muscle function,

memory, and dementia through increased cholinergic signaling while limiting the deleterious metabolic consequences secondary to currently available therapies.

# Materials and methods

**Key resources table**

| Reagent type (species) or resource | Designation | Source or reference | Identifiers | Additional information |
|---|---|---|---|---|
| Strain, strain background (*Escherichia coli*) | Mix & Go Competent Cells-Strain HB10 | Zymo Research | T3013 | |
| Genetic reagent (*Mus musculus*) | *Tigar$^{-/-}$* (TKO) (C57BL/6J) | PMCID:PMC5961042 | MMRRC, Cat# 063577-UCD | |
| Genetic reagent (*M. musculus*) | C57BL/6N-Tigar$^{tm1b(EUCOMM)Wtsi}$/Wtsi | Welcome Trust Sanger Institute (Hinxton Cambridge, UK) | IMSR Cat# EM:09836; RRID:IMSR_ EM:09836 | |
| Genetic reagent (*M. musculus*) | *Myl1$^{Cre}$*: Myl1$^{tm1(cre)Sjb}$/J | The Jackson Laboratory | RRID:IMSR_ JAX:024713 | |
| Genetic reagent (*M. musculus*) | *Chat$^{Cre}$*: B6J.129S6-Chat$^{tm2(cre)Lowl}$/MwarJ | The Jackson Laboratory | RRID:IMSR_ JAX:028861 | |
| Genetic reagent (*M. musculus*) | *Tigar$^{fl/fl}$* (C57BL/6J) | This study | N/A | Described in 'Mice models' |
| Genetic reagent (*M. musculus*) | *Tigar$^{fl/fl}$Adipoq$^{Cre}$* (C57BL/6J) | This study | N/A | Described in 'Mice models' |
| Genetic reagent (*M. musculus*) | *Tigar$^{fl/fl}$Myl1$^{Cre}$* (C57BL/6J) | This study | N/A | Described in 'Mice models' |
| Genetic reagent (*M. musculus*) | *Tigar$^{fl/fl}$/Chat$^{Cre}$* (C57BL/6J) | This study | N/A | Described in 'Mice models' |
| Genetic reagent (*M. musculus*) | *Ucp1$^{-/-}$* (UKO) (C57BL/6J) | Laboratory of Victor L. Schuster | Albert Einstein College of Medicine | Described in 'Mice models' |
| Genetic reagent (*M. musculus*) | *Tigar$^{-/-}$/Ucp1$^{-/-}$* (UTKO) (C57BL/6J) | This study | N/A | Described in 'Mice models' |
| Cell line (*Homo sapiens*) | SH-SY5Y neuroblastoma | ATCC | ATCC CRL-2266 | |
| Cell line (*Homo sapiens*) | SH-SY5Y Control neuroblastoma | This study | N/A | Materials and methods |
| Cell line (*Homo sapiens*) | SH-SY5Y TIGAR knockout neuroblastoma | This study | N/A | Materials and methods |
| Antibody | Anti-c-Fos (9F6) (rabbit monoclonal) | Cell Signaling Technology | Cat# 2250S; RRID:AB_2247211 | (1:1000) |
| Antibody | Rabbit anti-AMPKα | Cell Signaling Technology | Cat# 2532S; RRID:AB_330331 | (1:1000) |
| Antibody | Rabbit anti-AMPKα phospho (Thr172) | Cell Signaling Technology | Cat# 2535S; RRID:AB_331250 | (1:1000) |
| Antibody | Anti-UCP1 (EPR20381) (rabbit monoclonal) | Abcam | Cat# ab209483; RRID:AB_2722676 | (1:2000) |
| Antibody | Anti-TIGAR (rabbit polyclonal) | Millipore/Sigma | Cat# AB10545; RRID:AB_10807181 | (1:1000) |
| Antibody | Anti-VChAT (IF) (goat polyclonal) | ImmunoStar | Cat# 24286; RRID:AB_572269 | (1:100) |
| Antibody | Anti-VChAT (rabbit polyclonal) | Thermo Fisher Scientific | Cat# PA5-95404; RRID:AB_2807207 | (1:1000) |

*Continued on next page*

*Continued*

| Reagent type (species) or resource | Designation | Source or reference | Identifiers | Additional information |
|---|---|---|---|---|
| Antibody | Anti-ChAT (rabbit monoclonal) | Millipore Sigma | Cat# AB144P; RRID:AB_2079751 | (1:1000) |
| Antibody | Rabbit anti-choline transporter | Thermo Fisher Scientific | Cat# PA5-77385; RRID:AB_2736619 | (1:1000) |
| Antibody | Anti-TIGAR (E-2) (mouse monoclonal) | Santa Cruz Biotechnology | Cat# sc-166290; RRID:AB_2066582 | (1:250) |
| Antibody | Anti-TIGAR (E-2) Alexa Fluor 488 (IF) (mouse monoclonal) | Santa Cruz Biotechnology | Cat# sc-166290 AF488 | (1:100) |
| Antibody | Alexa Fluor 594 donkey anti-goat IgG (H+L) | Thermo Fisher Scientific | Cat# A-11058; RRID:AB_2534105 | (1:1000) |
| Antibody | Anti-β-actin (C4) (mouse monoclonal) | Santa Cruz Biotechnology | Cat# sc-47778; RRID:AB_626632 | (1:300) |
| Antibody | Anti-vinculin (7F9) (mouse monoclonal) | Santa Cruz Biotechnology | Cat# sc-73614; RRID:AB 1131294 | (1:300) |
| Antibody | Goat anti-mouse IgG-HRP | Santa Cruz Biotechnology | Cat# sc-2005 | (1:5000) |
| Antibody | Goat anti-rabbit IgG (H+L), HRP | Fisher Scientific | Cat# PI32460 | (1:5000) |
| Antibody | Goat anti-rabbit 800CW | LI-COR | Cat# 926-32211 | (1:6000) |
| Antibody | Goat anti-mouse 800CW | LI-COR | Cat# 926-32210 | (1:6000) |
| Antibody | Donkey anti-goat 800CW | LI-COR | Cat# 926-32214 | (1:6000) |
| Recombinant DNA reagent | 3xsgRNA/Cas9 all-in-one expression clone targeting TIGAR (NM_020375.2) | GeneCopoeia | Cat# HCP-215394-CG04-3 | |
| Recombinant DNA reagent | 3xsgRNA/Cas9 all-in-one expression clone scrambled sgRNA (control) | GeneCopoeia | Cat# CCPCTR01-CG04-B | |
| Sequence-based reagent | *Tigar*<fl/fl> genotyping forward | This study | IMSR Cat# HAR:7395; RRID:IMSR_HAR:7395 (9514-9534) | 5'-AGGGGGTTG CACCTCTATCTC |
| Sequence-based reagent | *Tigar*<fl/fl> genotyping reverse | This study | IMSR Cat# HAR:7395; RRID:IMSR_HAR:7395 (16933-16954) | 5'-CACACAAG AAGGAAGCTGTTGG |
| Commercial assay or kit | Direct-zol RNA MiniPrep with 200 ml TRI Reagent | Zymo Research | Cat# R2053 | |
| Commercial assay or kit | QuickDetect acetylcholine (ACh) (mouse) ELISA Kit | BioVision Inc | Cat# E4453-100 | |
| Commercial assay or kit | PowerPrep HP Plasmid Maxiprep Kits with Prefilters | Origene | NP100025 | |
| Commercial assay or kit | Ceria Stabilized Zirconium Oxide Beads | MidSci | Cat# GB01 | |
| Commercial assay or kit | ProLong Gold Antifade Reagent with DAPI | Cell Signaling Technology | Cat# 8961 | |
| Commercial assay or kit | SuperScript IV VILO Master Mix | Thermo Fisher Scientific Invitrogen | Cat# 11756050 | |

*Continued on next page*

*Continued*

| Reagent type (species) or resource | Designation | Source or reference | Identifiers | Additional information |
|---|---|---|---|---|
| Commercial assay or kit | TaqMan Universal Master Mix II, no UNG | Thermo Fisher Scientific Invitrogen | Cat# 4440049 | |
| Chemical compound, drug | Tubocurarine hydrochloride pentahydrate | Millipore/Sigma | Cat# T2379; CAS: 6989-98-6 | |
| Chemical compound, drug | Cyclopiazonic acid from *Penicillium cyclopium* | Millipore/Sigma | Cat# C1530; CAS: 18172-33-3 | |
| Chemical compound, drug | D-Glucose-1,2-$^{13}C_2$ | Millipore/Sigma | Cat# 453188; CAS: 138079-87-5 | |
| Chemical compound, drug | D-Glucose-$^{13}C_6$ | Millipore/Sigma | Cat# 389374; CAS: 110187-42-3 | |
| Chemical compound, drug | MG-132 | Millipore/Sigma | Cat# 474787; CAS: 133407-82-6 | |
| Chemical compound, drug | ALLN | Millipore/Sigma | Cat# 208719-25MG; CAS:110044-82-1 | |
| Chemical compound, drug | RIPA lysis buffer | Santa Cruz Biotechnology | Cat# sc-24948 | |
| Chemical compound, drug | Halt Protease and Phosphatase Inhibitor Cocktail | Thermo Fisher Scientific | Cat# 78442 | |
| Chemical compound, drug | Pierce Protein-Free T20 (TBS) Blocking Buffer | Thermo Fisher Scientific | Cat# 37571 | |
| Chemical compound, drug | SuperSignal West Pico Chemiluminescent Substrate | Thermo Fisher Scientific | Cat# 34080 | |
| Chemical compound, drug | SuperSignal West Femto Maximum Sensitivity Substrate | Thermo Fisher Scientific | Cat# 34096 | |
| Chemical compound, drug | 4% paraformaldehyde solution | Electron Microscopy Sciences | Cat# RT157-4 | |
| Chemical compound, drug | Vectashield Antifade Mounting Medium with DAPI | Vector Laboratories, Inc | Cat# H1200 | |
| Chemical compound, drug | DMEM medium without glucose, glutamine, and sodium pyruvate | Biological Industries USA | Cat# 01-0571-1A | |
| Chemical compound, drug | DMEM medium no glucose, glutamine, and sodium pyruvate | Biological Industries USA | Cat# 01-0571-1A | |
| Software, algorithm | Odyssey Application Software 2.0 | LI-COR | N/A | |
| Software, algorithm | GraphPad Prism 8 | GraphPad Software | http://www.graphpad.com/scientific-software/prism/ | |
| Software, algorithm | RQ Manager 1.2.1. | Thermo Fisher Scientific Applied Biosystems | N/A | |
| Software, algorithm | LD50 Calculator | AAT Bioquest | https://www.aatbio.com/tools/ld50-calculator | |
| Software, algorithm | Velocity software | Velocity Software, Inc | https://www.velocitysoftware.com/ | |
| Software, algorithm | MassHunter | Agilent Technologies | Version: B08.00 | |

*Continued on next page*

*Continued*

| Reagent type (species) or resource | Designation | Source or reference | Identifiers | Additional information |
|---|---|---|---|---|
| Software, algorithm | MultiQuant | ABSciex | Version: 3.0.2 | |
| Software, algorithm | LabChart 8.1 for Macintosh | ADInstruments | https://www.adinstruments.com/products/labchart | |
| Other | Thermalert Model TH-8 Temperature Monitor | Physitemp Instruments LLC. | Thermalert Model TH-8 | |
| Other | HD-X11 telemetric transponder probe | DSI, Harvard Bioscience, Inc | https://www.datasci.com/products/implantable-telemetry/small-animal-telemetry | |
| Other | DSI Implantable Telemetry | DSI, Harvard Bioscience, Inc | https://www.datasci.com/products/implantable-telemetry/small-animal-telemetry | |
| Other | GC-MS instrument | Agilent Technologies | Model: 5975C | |
| Other | DB-5MS column | Agilent Technologies | Cat# 122-5532G | |
| Other | LC-MS/MS instrument | ABSciex | Model: 6500+ | |
| Other | ZIC-pHILIC column | Merck | Cat# 1.50460.0001 | |

## Resource availability

### Lead contact

Further information and requests for resources and reagents should be directed to and will be fulfilled by the lead contact, Jeffrey E Pessin (Jeffrey.pessin@einsteinmed.edu).

### Materials availability

This study did not generate new unique reagents. SH-SY5Y neuroblastoma both scrambled and TKO cell lines are available upon request from the lead contact. Mouse lines generated in this study are available from the lead contact, although there are timeline restrictions to the availability of the mice due to the labor and space limitations of the laboratory staff and the facility of the Institution for Animal Studies at Albert Einstein College of Medicine.

## Experimental models and subject details

### Mice models

All studies were performed in accordance with protocols approved by the Einstein Institutional Animal Care and Use Committee. TKO mice were used as described in our previous paper (*Tang et al., 2018*). To establish conditional knockout mice of *Tigar* gene, we purchased C57BL/6N-^TIGARtm1b(EUCOMM)Wtsi/Wtsi mice from the Welcome Trust Sanger Institute (Hinxton Cambridge, UK). Briefly, loxP sites were generated to delete exon 3 of *Tigar*, chimeric mice were mated with C57BL/6mice, and germ-line transmission was confirmed by PCR and Southern blotting. The progeny crossed sequentially with *FLPe* transgenic mice to remove *lacZ* and *neo* genes. The primer pairs TIGARCKO-5FRT-FW: TTGGGATCCCCTAGTTTGTG and TIGARCKO-5FRT-RV: AACTCAGCCTTGAGCCTCTG were used to confirm the deletion of the *lacZ* and neomycin resistance genes. The primer pairs TIGARCKO-3loxp-FW: GAGAAGAGACCCCCTGGAAC and TIGARCKO-3loxp-RV: TTCCGGCCAAACAGACTTAC were used to confirm the *Tigar^fl/fl* mice. Mice were backcrossed more than 10 generations to the C57BL/6J mice. The primers for subsequent genotyping for *Tigar^fl/fl* are listed in the Key resources table, and the mouse tail PCR (touchdown cycling protocol, The Jackson Laboratory) results in 538 bp for mutant and 346 bp for the WT mice.

Adipocyte-specific TKO (*Tigar^fl/fl*/*Adipoq^Cre*) mice were produced by mating *Tigar^fl/fl* mice with *Adiponectin-Cre* (*Adipoq^Cre*) mice, as described previously (*Feng et al., 2018*). Skeletal muscle-specific

TKO mice were produced by mating the *Tigar<sup><fl/fl></sup>* mice with skeletal muscle-specific myosin light polypeptide 1 Cre (*Myl1<sup>Cre</sup>*) mice purchased from The Jackson Laboratory (stock number 024713). chTKO mice were generated by mating *Tigar<sup>fl/fl</sup>* mice with *Chat<sup>Cre</sup>* mice purchased from The Jackson Laboratory (stock number 028861). UKO mice, which were C57BL/6J background, were a gift from Dr. Victor Schuster (Albert Einstein College of Medicine). UTKO mice were generated by breeding UKO mice with TKO, which resulted in the production of WT, TKO, UKO, and UTKO mice.

## Mouse husbandry and rectal temperature measurement

Mice were housed in a facility equipped with a 12 hr light/dark cycle. Animals were fed a normal chow diet that contains 62.3% (kcal) carbohydrates, 24.5% protein, and 13.1% fat (5053, LabDiet). The mice at 12–16 weeks of age were used for rectal temperature measurement using TH-8 Thermometer (Physitemp Instruments, Clifton, NJ) as per the manufacturer's instructions. For cold exposure, the mouse was individually put into a pre-chilled cage with bedding in a cold chamber (4°C). For tissue collection, the mice were killed, and the tissues were collected and snap-frozen in liquid nitrogen and stored in a −80°C freezer. All studies were approved by and performed in compliance with the guidelines of the Albert Einstein College of Medicine Institutional Animal Care and Use Committee.

## Blood plasma collection

Mice were briefly anesthetized by isoflurane, and the blood sample was collected from mouse orbital sinus into microtube with EDTA (Microvette 500 K3E, SARSTEDT, Germany) on ice. The plasma sample was collected by centrifugation (1000 × *g*, 30 min) and was stored at –80°C.

## Gastrocnemius and quadricep skeletal muscle samples isolation

Gastrocnemius and quadricep white muscles were collected and snap-frozen in liquid nitrogen and stored at −80°C. The whole piece of the muscle was ground using a mortar-pestle in liquid nitrogen, and the muscle powder (30 mg) was used for metabolomics analysis.

## Tubocurare and CPA treatments

Tubocurare (T2379, Sigma-Aldrich) was given as an intraperitoneal injection in a total volume of 100 µl in saline solution reaching at various doses and times indicated in the figure legends, and the median lethal dose (LD$_{50}$) was calculated with LD$_{50}$ Calculator, an online tool from AAT Bioquest (Sunnyvale, CA). CPA (C1530, Sigma-Aldrich) was given as an intraperitoneal injection in a total volume of 100 µl in saline solution reaching a dose of 10 mg/kg body weight for the time indicated. In both cases, the same volume of 100 µl of saline were injected as vehicle controls.

## TIGAR knockout in human SH-SY5Y neuroblastoma cells

Human SH-SY5Y neuroblastoma cell line was purchased from ATCC (ATCC CRL-2266). The SH-SY5Y cells were cultured in Dulbecco's modified Eagle's medium (DMEM) supplemented with 10% fetal bovine serum and 1× penicillin-streptomycin. Cell lines were maintained in a 5% $CO_2$ incubator at 37°C and were routinely tested to exclude *Mycoplasma* contamination.

The CRISPR *3xsgRNA/Cas9* all-in-one expression clone targeting *Tigar* (NM_020375.2) (Cat# HCP215394-CG04-3) and scrambled sgRNA control for pCRISPR-CG04 (Cat# CCPCTR01-CG04-B) were purchased from GeneCopoeia (Rockville, MD). The plasmid DNAs were transformed and amplified in Mix & Go Competent Cells-Strain HB 101 (Zymo Research, Cat# T3013, Irvine, CA) and purified using PowerPrep HP Plasmid Maxiprep kits with prefilters (Origene, Rockville, MD). The SH-SY5Y neuroblastoma cells were transfected with 10 µg of plasmid DNAs when 60% confluence in 100 mm dishes using transfection reagents GenJet II (SignaGen Laboratories, Rockville, MD) as per the manufacturer's instructions. The transfected cells were cultured for 48 hr and collected in 500 µl of MACS buffer (phosphate-buffered saline [PBS-containing EDTA]) containing 1% fetal bovine serum in Falcon 5 ml polystyrene round-bottom tube filtered with a cell-strainer cap. The GFP-positive cells were selected using FACS GFP sorting and were cultured and proliferated in 10% FBS DMEM. The cell descent was collected, and cell lysate was immunoblotted with a TIGAR antibody.

## Method details

### In vivo $^{18}$O isotopic labeling of cellular phosphoryls of quardricep white muscle of the mice

WT and TKO male mice (n = 3–4 per condition) were kept at room temperature or shifted to 4°C for 1 hr and then given an oral gavage of 0.3 ml pure $H_2^{18}O$ for 10 min at 4°C and an IP injection of 1 ml saline $H_2^{18}O$ (0.9% NaCl in $^{18}$O water) for another 10 min at 4°C. The dosing and timing of this protocol were selected from pilot studies to ensure steady-state labeling of the ATP pool. The quadricep muscles were freeze-clamped with liquid $N_2$ and extracts prepared for mass spectroscopic analyses for the identification of the position of $^{18}$O into the phosphates of ATP, which can be extrapolated to ADP and AMP (*Nemutlu et al., 2015*).

### Purification and isotopic analysis of $^{18}$O-labeled cellular phosphoryls

Quadricep muscles were freeze-clamped and pulverized in mortar with liquid nitrogen, and extracted in 80% methanol solution. The samples were centrifuged at 10,500 rpm for 10 min at 4°C to precipitate proteins. The supernatants were transferred to an LCMS sampling vial pending for injection. Metabolite separation was performed on a ZIC-pHILIC column (Merck). Data was collected with ABSciex 6500 + QTRAP MS/MS with a multiple reaction monitoring (MRM) mode. The ions monitored are listed as follows:

| Metabolites | Isotope labels | Q1 | Q3 |
|---|---|---|---|
| AMP | M0 | 346 | 79 |
| AMP | M2 | 348 | 81 |
| AMP | M4 | 250 | 83 |
| ADP | M0 | 426 | 79 |
| ADP | M2 | 428 | 81 |
| ADP | M4 | 430 | 83 |
| ADP | M0 | 426 | 159 |
| ADP | M2 | 428 | 161 |
| ADP | M4 | 430 | 163 |
| ADP | M6 | 432 | 165 |
| ATP | M0 | 506 | 79 |
| ATP | M2 | 508 | 81 |
| ATP | M4 | 510 | 83 |
| ATP | M0 | 506 | 159 |
| ATP | M2 | 508 | 161 |
| ATP | M4 | 510 | 163 |
| ATP | M6 | 512 | 165 |
| ATP | M0 | 506 | 239 |
| ATP | M2 | 508 | 241 |
| ATP | M4 | 510 | 243 |
| ATP | M6 | 512 | 245 |
| CP | M0 | 210 | 79 |
| CP | M2 | 212 | 81 |
| CP | M4 | 214 | 83 |

The $^{18}$O concentration in mouse body was determined with GC-MS according to the MMPC protocol.

Briefly, 10 µl of plasma from the matched quadricep muscle isolated mice were added to 2 µl of 5 N NaOH and 4 µl of % acetone in acetonitrile to allow for $^{16}$O to $^{18}$O exchange (*Aksnes et al., 1966*).

The samples were kept at room temperature overnight (at least 10 hr). After chloroform extraction, the samples were analyzed on an Agilent GC-MS system with EI mode. Selected ion monitoring (SIM scan with ions of 58, 59, and 60) was used for the detection. A linear curve of different percentages (from 0% to 10%) of $H_2^{18}O$ in water was prepared along with the samples. $^{18}O$ enrichment was calculated with the linear curve. The enrichment of $^{18}O$ in phosphates in skeletal muscle (see below) was normalized to the $^{18}O$ water enrichment in the plasma.

## Stable isotope enrichment and acetylcholine analyses in culture media of SH-SY5Y control and TKO neuroblastoma cells

The SH-SY5Y neuroblastoma cells were cultured in 10 cm dishes until confluent and the culture media were removed, and the cells were washed with DMEM medium (Biological Industries, Boston, Cat# 01-057-1A, without glucose, glutamine, and pyruvate). Ten ml of DMEM containing either 10 mM D-glucose-1,2-$^{13}C_2$ (Sigma-Aldrich 453188-1G) or 10 mM D-glucose-$^{13}C6$ (U-[$^{13}C$]-glucose) and 10 mM D-glucose (unlabeled:labeled = 1:1), 1 mM glutamine, without pyruvate, was added to the cultured cells, and medium was collected at 4, 8, and 24 hr time points. The medium was centrifuged at 1000 × $g$ for 20 min at 4°C and stored at –80°C. The plates at 24 hr time point were put on ice and gently rinsed with 1.5 ml of ice-cold 150 mM ammonium acetate. The cells were scraped and transferred into a 2 ml screw cap tube (Cat# 02682558, Thermo Fisher). The cell pellets were collected by 1000 × $g$ centrifugation at 4°C for 2 min and stored at –80°C for cellular acetyl-CoA and acetyl carnitine measurement.

Following thawing on ice, the medium was subjected to an ethyl chloroformate (ECF) derivatization according to our developed procedure with a minor modification (*Xie et al., 2007*). Briefly, a volume of 700 µl of cell culture media was added to 500 µl of ethanol:pyridine = 4:1, and 100 µl of ECF. After a brief vortex, the samples were sonicated for 1 min. The derivatized metabolites were extracted with 300 µl of chloroform and injected 1 µl into the GC-MS system in splitless mode (GC-MS, Agilent, USA). Helium was used as a carrier gas at a consistent flow of 1 ml/min. The injection temperature was set at 260°C, and metabolite separation was performed on an Agilent DB-17MS column. The oven program started at 60°C for 0.5 min and rose to 100°C at a rate of 20°C/min, and then to 300°C at a rate of 10°C/min, and then maintained at 300°C for 4.5 min. The data was analyzed with MassHunter Quantitative Analysis software (Agilent).

The m0 glutamate fragment $m/z$ of 128 contains three carbons (carbon 2–4) out of the five carbons, while fragment $m/z$ of 202 contains four carbons (carbon 2–5) from the whole glutamate molecule. M1 of glutamate at $m/z$ 129 reflects the presence of $^{13}C$ at the fourth carbon position. This reflects acetyl units originating from glucose-> pyruvate entering through PDH for eventual oxidation (glycolytic flux for energy generation). M2 of glutamate $m/z$ 130 reflects the presence of $^{13}C$ at the second and third carbon positions, and reflects pyruvate entering the oxaloacetate pool or through PC (*Madhu et al., 2020*; *Silagi et al., 2020*). M2 of glutamate at 204 is the overall TCA cycle flux. Those two $^{13}C$ come from either pyruvate entering from PC or PDH. For lactate, the $m/z$ of M0 to M2 for carbon 2 and 3 positions will be 145–147 (*Madhu et al., 2020*). A pooled quality control (QC) sample was injected six times for coefficient of variation (CV) determination. Metabolites with CVs lower than 30% were included for quantification while loss with CVs greater than 30% were considered indeterminant. These data were obtained using a Sciex 6500 + QTRAP with ACE PFP and Merck ZIC-pHILIC columns as described previously (*Hoshino et al., 2012*).

## Acetylcholine enrichment in culture media of SH-SY5Y control and TKO neuroblastoma cells

The cell culture medium samples were thawed on ice and 200 µl was added to 800 µl of acetonitrile plus 10 µl of 10 ng/ml acetylecholine_d9 (deuterium 9 as an internal control). The samples were vortex, centrifuged at 14,000 rpm for 10 min, and supernatants were transferred into glass vials. The samples were analyzed with ABSciex 6500 + with ACE PFP column. The MRMs for m0, m1, m2 for acetylcholine and the internal standard are as given in the following table. Data was processed with multiquanta software (ABSciex). The enrichment was calculated after subtracting from the background of the non-labeled treatment samples.

## MRM table

| | | Q1 mass (Da) | Q3 mass (Da) |
|---|---|---|---|
| 1 | Acetylcholine | 146.3 | 87.0 |
| 2 | Acetylcholine_m1 | 147.3 | 88.0 |
| 3 | Acetylcholine_m2 | 148.3 | 89.0 |
| 4 | (d9) Acetylcholine | 155.3 | 87.0 |

## Cellular acetyl-CoA and acetyl-carnitine measeurement in SH-SY5Y control and TKO neuroblastoma cells

The cell pellet samples were added with 80% methanol with internal standards (malonyl CoA_$^{13}$C3, and acetyl-carnitine_d3) extraction with three cycles of frozen (liquid nitrogen)-thaw (sonication). The samples were vortex, centrifuged at 21,000 × $g$ for 10 min, and the supernatants were transferred into glass vials. The samples were analyzed with ABSciex 6500 + iHILIC-p column (HILICON). Data was processed with multiquanta software (ABSciex).

## Total RNA extraction and quantitative RT-PCR

Cellular and tissue total RNA was extracted using TRI reagent and Direct-zol RNA MiniPrep Plus kit (Zymo Research). First-strand cDNA was synthesized using the SuperScript IV VILO cDNA synthesis kit (Thermo Fisher Scientific). TaqMan RT-PCR was performed for the measurement of mRNA using the $\Delta\Delta C_t$ method. Gene expression was adjusted by comparison with *Rpl7* expression. The quantitative RT-PCR results were analyzed with RG Manager version 1.2.1 (Applied Biosystems). Primer-probe mixtures for the genes *Ucp1, Atp2a1 (SERCA1), Atp2a2 (SERCA2), Sln (sarcolipin), Mrln (myoregulin), Chrna1, Chrnb1, Chrnd, Chrne,* and *Chrng* are listed as follows:

| Gene name | Gene name | TaqMan assay ID |
|---|---|---|
| Uncoupling protein 1 (mitochondrial, proton carrier) | *Ucp1* | Mm01244861_m1 |
| ATPase, Ca++ transporting, cardiac muscle, fast twitch 1 (SERCA1) | *Atp2a1* | Mm01275320_m1 |
| ATPase, Ca++ transporting, cardiac muscle, slow twitch 2 (SERCA2) | *Atp2a2* | Mm01201431_m1 |
| Sarcolipin | *Sln* | Mm00481536_m1 |
| Myoregulin | *Mrln* | Mm01175781_m1 |
| Cholinergic receptor, nicotinic, alpha polypeptide 1 (muscle) | *Chrna1* | Mm00431629_m1 |
| Cholinergic receptor, nicotinic, beta polypeptide 1 (muscle) | *Chrnb1* | Mm00680412_m1 |
| Cholinergic receptor, nicotinic, delta polypeptide | *Chrnd* | Mm00445545_m1 |
| Cholinergic receptor, nicotinic, epsilon polypeptide | *Chrne* | Mm00437411_m1 |
| Cholinergic receptor, nicotinic, gamma polypeptide | *Chrng* | Mm00437419_m1 |

Primer-probe mixture for *Rpl7* was customized, and other primer-probe mixtures were obtained from Thermo Fisher Applied Biosystems.

## Immunoblotting

Commercial primary antibodies were purchased for the detection of TIGAR (Cat# AB10545, Millipore/Sigma; Cat# sc-677290, Santa Cruz), ChAT (Cat# AB144P, Millipore/Sigma), VAChT (Cat# 24286, ImmunoStar), ChT (Cat# PA5-77385, Thermo Fisher), c-fos (2250S, Cell Signaling; MA5-15055, Invitrogen), phosphor-c-Fos$^{Ser32}$ (5348S, Cell Signaling), AMPKα (2532S, Cell Signaling), phospho-threonine 172 AMPKα (2535S, Cell Signaling), UCP1 (ab209483, Abcam), vinculin (sc-73614), and β-actin (sc-47778, Santa Cruz). Samples were prepared from culture cells (washed by cold PBS) or tissues by homogenization with a radioimmune precipitation assay lysis buffer (sc-24948, Santa Cruz Biotechnology) containing Halt protease and phosphatase inhibitor mixture (Cat# 78442, Thermo Fisher Scientific), 100 μM MG132, and 100 μM ALLN (EMD Millipore, Darmstadt, Germany) using Ceria stabilized zirconium oxide beads (MidSci, Valley Park, MO). Homogenates were centrifuged for 15 min at 21,000

× g at 4°C, and supernatants were collected for the protein assay using the BCA method. Protein samples were separated by SurePAGE (Bis-Tris) precasted gel (GenScript, Piscataway, NJ) and transferred to nitrocellulose membrane using iBlot Blotting System (Thermo Fisher Scientific). The immunoblot membrane was blocked with Pierce Protein-Free T20 (TBS) blocking buffer (product no. 37571, Thermo Fisher Scientific) and incubated with the first antibody indicated in the blocking buffer. Blots were washed in TBS with Tween 20 (TBST) and incubated with either IRDye 800CW secondary antibody (LI-COR, Lincoln, NE) or horseradish peroxidase-conjugated secondary antibody in blocking buffer. The membrane was washed with TBST and visualized either by the Odyssey Imaging System (LI-COR) or enhanced chemiluminescence (ECL) (Thermo Fisher Scientific Pierce) method. ImageJ was used to quantify protein bands on the membrane.

### Fluorescence imaging of nicotinic acetylcholine receptors clustering at neuromuscular junctions

Whole EDL muscles were fixed 30 min with 4% w/v paraformaldehyde after dissection. After PBS washing 5 min × three times, EDL muscles were incubated with 1:1000 dilution of Alexa Fluor 488 conjugate α-Bungarotoxin (Invitrogen B13422) for 2 hr at room temperature, and mounted onto 35 mm glass-bottom dish (MatTek Corporation, Cat# P35G020C.S) with Vectashield Antifade Mounting Medium with DAPI (Cat# H1200, Vector Laboratories, Inc, Burlingame, CA). The NMJ was visualized using a z-series projection on a confocal microscope (Leica SP8) with ×40 objective. Velocity software was used for image analysis.

### Immunofluorescence imaging of SCG

The *Chat^Cre* and chTKO male mice (14 weeks old) were euthanized and the fresh SCG tissues were removed (*Figure 4—figure supplement 1*) and embedded in optimal cutting temperature compound. The frozen tissue cross-sections (10 μm) were blocked with 3% bovine serum albumin in PBS for 60 min at room temperature. For VChAT and TIGAR immunofluorescence staining, the SCG section was incubated with the mouse monoclonal anti-TIGAR (E-2) Alexa Fluor 488 (sc-166290 AF488, Santa Cruz Biotechnology) and goat polyclonal anti-VChAT (Cat# 24286, ImmunoStar, Hudson, WI) for 2 hr at room temperature. After washing PBS 5 min × three times, Alexa Fluor 594 donkey anti-goat IgG (H+L) (1:1000, Cat# A-11058, Thermo Fisher Scientific) was added to the sections for 30 min at room temperature. The slides were mounted on coverslips with Vectashield Antifade Mounting Medium with DAPI (Cat# H1200, Vector Laboratories, Inc) and visualized using a z-series projection on a confocal microscope (Leica SP8) with ×40 objective.

### Synchronized measurement of core body temperature, blood pressure, heart beating rate, and electromyography

*Chat^Cre* and chTKO mice were anesthetized with isoflurane and implanted with DSI HD-X11 telemetric transponder probes according to the manufacturer's guidelines, with the blood pressure probe placed in the descending aorta, the EMG leads run subcutaneously to the neck muscles bilaterally, and the core temperature probe placed intraperitoneally. Animals were returned to a standard temperature housing room maintained at 21–22°C on a 12 hr/12 hr light:dark cycle, lights on at 0700. Mice were individually housed on cob bedding in standard polycarbonate home cages that were placed on telemetric DSI platforms to continuously monitor arterial blood pressure, heart rate, core body temperature, and neck EMG muscle activity from the implanted DSI XD-11 probes using LabChart 8.1 software for Macintosh. After 1 week recovery from surgery, animals' home cage bedding was removed and replaced by stainless steel mesh flooring 1 inch above the base of the cage. Food but not water was then removed from the home cage, and animals in their home cages on DSI platforms were then placed in a 10' × 10' environmental cold room maintained at 4°C, 35% humidity for 1 hr from 10:00 to 11:00 am, and continuous measurements of heart rate, blood pressure, core temperature, and EMG activity continued. At the end of the 1 hr cold challenge, mice in their individual cages were then returned to the standard temperature housing room, and food and bedding were replaced.

## Mitochondrial OXPHOS efficiency and capacity

Respiration of permeabilized muscle fibers was performed as described previously (*Heden et al., 2017*; *Johnson et al., 2018*; *Heden et al., 2019*). Briefly, a small portion of freshly dissected red gastrocnemius muscle tissue was placed in 7.23 mM $K_2$EGTA, 2.77 mM Ca $K_2$EGTA, 20 mM imidazole, 20 mM taurine, 5.7 mM ATP, 14.3 mM phosphocreatine, 6.56 mM $MgCl_2.6H_2O$, and 50 mM K-MES (pH 7.1). Fiber bundles were separated and permeabilized for 30 min at 4°C with saponin (30 µg/ml) and immediately washed in 105 mM K-MES, 30 mM KCl, 10 mM $K_2HPO_4$, 5 mM $MgCl_2.6H_2O$, BSA (0.5 mg/ml), and 1 mM EGTA (pH 7.4) for 15 min. After washing, high-resolution respiration rates were measured using an OROBOROS Oxygraph-2k.

## Muscle contraction/relaxation

The EDL muscle was dissected and tied into the Horizontal Tissue Bath System from Aurora Scientific Inc as described previously (*Ferrara et al., 2018*). The muscles were stimulated with a 20 V twitch train and stretched until optimal length for force production was reached. Muscles were then allowed to equilibrate for 5 min. For force-frequency curves, muscles were stimulated with frequencies ranging from 10 to 200 Hz (0.1 ms pulse, 330 ms train, and 2 min between trains). Forces produced by electrically stimulated muscle contractions were recorded in real time via a force transducer (model 400 A, Aurora Scientific Inc). The specific force was calculated using the cross-sectional area of the muscle tissue (millinewton per square millimeter), as estimated from the weight and length of the muscle.

## Gastrocnemius muscle acetylcholine assay

Fresh gastrocnemius muscle was collected and snap-frozen in liquid nitrogen, and the whole piece of the muscle was ground to powder using a mortar-pestle in liquid nitrogen. The muscle powder was then homogenized in PBS (pH 7.4) using Ceria stabilized zirconium oxide beads (MidSci) and the homogenate was centrifuged for 20 min at 3000 rpm at 4°C. The supernatant was collected, and the protein assay was performed using the Pierce BCA Protein Assay kit (Cat# 23225, Thermo Fisher Scientific). The supernatants from different samples were adjusted to the same protein concentration of protein, and the same volume of the supernatants was subjected to acetylcholine assay using QuickDetect acetylcholine (ACh) (mouse) ELISA Kit (Cat# E4453-100, Biovision, Milpitas, CA).

## Body composition and calorimetry

Male WT C57BL/6J and TKO mice (n = 4) were internally transferred to the Animal Energy Balance Phenotyping Core of the New York Obesity Research Center at the Albert Einstein College of Medicine. In vivo body composition was determined by magnetic resonance spectroscopy using an ECHOMRI instrument (Echo Medical Systems). The mice were then individually housed in metabolic chambers maintained at 21–22°C on a 12 hr light/12 dark cycle with lights on at 7 am and were provided with a nutritionally complete standard powdered diet (NIH 07) and tap water ad libitum for five consecutive days. Metabolic and behavioral measurements (oxygen consumption, food intake, locomotor activity) were obtained continuously using a CLAMS (Columbus Instruments) open-circuit indirect calorimetry system (*Li et al., 2011*).

## Measurement of OCR in SH-SY5Y control and TKO neuroblastoma cells

Real-time OCR was simultaneously assessed in SH-SY5Y control and TKO neuroblastoma cells using a Seahorse Bioscience XF96 extracellular flux analyzer (*Zhang and Zhang, 2019*). Three batches of control CTL (10 wells/batch) and TKO (10 wells/batch) cells, respectively, were performed for the OCR assay. The cell suspension of $5 \times 10^4$ in 50 µl of DMEM with 10% fetal bovine serum was seeded into a well of the XF 96-well microplates to reach confluence growing an even monolayer. 16 hr later, 200 µl/well of medium were added. The cells were at 95–100% confluence and subjected to Seahorse analysis at the Stable Isotope & Metabolomics Core Facility at the Albert Einstein College of Medicine. Briefly, cells were rinsed and incubated with XF base medium (Agilent, 102353-100) supplemented with 25 mM glucose, 2 mM L-glutamine (Thermo Fisher Scientific, 25030081) and 1 mM sodium pyruvate (Thermo Fisher Scientific, 11360070), pH 7.4. After 45 min incubation in a $CO_2$-free incubator at 37°C, OCR was recorded at baseline, followed by sequential additions of 1 µM oligomycin (Sigma-Aldrich, 75351), 1 µM FCCP (Sigma-Aldrich, C2920), and 1 µM rotenone (Sigma-Aldrich, R8875). The

OCR was automatically recorded and calculated using the Seahorse XF24 software, and the data were exported into Prism 8.1 software for graph production and statistical analysis.

### Soleus and EDL muscles crossed section HE staining

Fresh soleus and EDL skeletal muscles from 4-month-old male WT and TKO mice were isolated and fixed in 10% formalin for 48 hr at 4°C. The muscle samples were then switched to 70% alcohol and subjected to H&E staining at the Histology and Comparative Pathology Core at the Albert Einstein College of Medicine. Briefly, fixed muscle samples were embedded in cassettes perpendicularly with paraffin. The block was sectioned 10 μm thick, and the sections were mounted on slides, H&E stained, and covered with clips. The H&E-stained images were captured with a digital microscope camera.

### Quantification and statistical analyses

Prism (8.1) GraphPad Software was used for data processing, analyses, and graph productions in the experiments. To analyze the differences of body temperature at different time points among two to four groups of the mice, the tabs of Grouped Analyses/two-way ANOVA (or mixed model)/multiple comparison were selected. The subtab 'Compare each cell mean with the other cell mean in that row' was chosen to show the 'Sidak's multiple comparison test,' which summarized the statistical test results. The number of independent experimental replications and the average with standard deviation are provided in the figure legends. Unpaired two-tailed p-value t-tests were used for the statistical tests between the two groups. The statistical analyses were made at significance levels as follows: ns, not statistically significant; $*p<0.5$; $**p<0.1$; $***p<0.001$, and $****p<0.0001$.

## Acknowledgements

We thank Dr. Yu Zhang at the Flow cytometry Core Facility (Albert Einstein College of Medicine) for SH-SY5Y cells FACS sorting. We thank Dr. Xue-liang Du at the Stable Isotope & Metabolomics Core Facility for Seahorse analyses. We also thank Dr. Victor Schuster (Albert Einstein College of Medicine) for the gift of UCP1 knockout mice. We thank Ms. Nicole Fernandez (Ph.D. candidate at the Albert Einstein College of Medicine) for her contribution to the production of Graphic Abstract. This study was supported by grants DK033823 and DK020541 from the National Institutes of Health, and an S10 SIG Award for the Sciex 6500 + QTRAP (1S10OD021798).

## Additional information

### Funding

| Funder | Grant reference number | Author |
| --- | --- | --- |
| National Institutes of Health | DK033823 | Jeffrey Pessin |
| National Institutes of Health | DK020541 | Jeffrey Pessin |
| S10 SIG Award for the Sciex 6500+QTRAP | 1S10OD021798 | Irwin J Kurland |

The funders had no role in study design, data collection and interpretation, or the decision to submit the work for publication.

### Author contributions

Yan Tang, Conceptualization, Data curation, Formal analysis, Funding acquisition, Investigation, Methodology, Project administration, Resources, Software, Supervision, Validation, Visualization, Writing – original draft, Writing – review and editing; Haihong Zong, Conceptualization, Data curation, Formal analysis, Funding acquisition, Investigation, Methodology, Resources, Software, Supervision, Validation, Visualization, Writing – original draft; Hyokjoon Kwon, Data curation, Formal analysis, Investigation, Methodology, Resources, Software, Validation, Visualization, Writing – original draft; Yunping Qiu, Data curation, Formal analysis, Investigation, Methodology,

Resources, Software, Supervision, Validation, Visualization, Writing – original draft; Jacob B Pessin, Formal analysis, Investigation, Methodology, Software, Validation, Writing – original draft; Licheng Wu, Data curation, Formal analysis, Investigation, Methodology, Resources, Software, Validation, Visualization; Katherine A Buddo, Ilya Boykov, Cameron A Schmidt, Chien-Te Lin, Data curation, Formal analysis, Investigation, Methodology, Resources, Software, Validation; P Darrell Neufer, Conceptualization, Data curation, Formal analysis, Investigation, Methodology, Resources, Software, Supervision, Validation, Visualization, Writing – original draft; Gary J Schwartz, Irwin J Kurland, Conceptualization, Data curation, Formal analysis, Investigation, Methodology, Resources, Software, Supervision, Validation, Writing – original draft; Jeffrey E Pessin, Conceptualization, Data curation, Formal analysis, Funding acquisition, Investigation, Methodology, Project administration, Resources, Supervision, Validation, Visualization, Writing – original draft, Writing – review and editing

## Author ORCIDs
Yan Tang ⬤ http://orcid.org/0000-0002-1377-422X
Jeffrey E Pessin ⬤ http://orcid.org/0000-0003-2041-2726

## Ethics
All studies were performed in accordance with protocols approved by the Einstein Institutional Animal Care and Use Committee. All of the animals were handled according to approved institutional animal care and use committee (IACUC) protocols (0000-1041, 0000-1061, and 0000-1389) of the Albert Einstein College of Medicine.

## Decision letter and Author response
Decision letter https://doi.org/10.7554/eLife.73360.sa1
Author response https://doi.org/10.7554/eLife.73360.sa2

---

# Additional files

## Supplementary files
• Transparent reporting form
• Source data 1. Figure source data and legends.

## Data availability
All data generated or analysed during this study are included in the manuscript and supporting file; Source Data files have been provided for Figures 1, 3, 4, 5, 7, Figure 1-figure supplement 1, 3, and 6.

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
