## [Editor Report]

The authors report results from timely studies, demonstrating the role of TIGAR in regulating thermoregulation. Deletion of TIGAR leads to resistance to cold-induced hypothermia. The results will be of wide interest.

---

## [Decision Letter]

**Decision letter after peer review:**

Thank you for submitting your article "Neuronal glucose metabolism sets cholinergic tone and controls thermo-regulated signaling at the neuromuscular junction" for consideration by *eLife*. Your article has been reviewed by 3 peer reviewers, and the evaluation has been overseen by a Reviewing Editor and Ma-Li Wong as the Senior Editor. The following individual involved in review of your submission has agreed to reveal their identity: Alan Saltiel (Reviewer #2).

Essential revisions:

1) Report body weights and body composition to exclude contributions from increased muscle mass. Metabolic cage data would eliminate alterations in physical activity, food consumption and basal energy expenditure as factors that protect from cold exposure.

2) Either provide data to support increased metabolic rates in the cell model (i.e. VO2, kinetic enrichment data, etc.) or reword the few discussion points that imply increased acetyl-CoA rates of production from glycolysis drive the cholinergic effect. As a side point, acetyl-CoA concentration could be measured in the cell model.

*Reviewer #1 (Recommendations for the authors):*

Can the authors explained what they mean by "since UCP1 protein levels lag mRNA expression there was no detectable UCP1 protein" in iWAT. Could it be that there just isn't detectable amount of protein there?

Authors indicate that both UKO and TKO mice were "both with C57BL/6J background (line 128) but methods indicate that UKO were on C57BL/6N background.

Loading controls are missing for Figure 1G.

Figure 4A's resolution is too low to be convincing.

Line 149: Figure S1G rather than 1G.

Statistical analysis: please describe two-way ANOVA multiple comparisons (what post-hoc tests were conducted when there was significant interactions/main effects?).

*Reviewer #2 (Recommendations for the authors):*

This interesting manuscript entitled "Neuronal glucose metabolism sets cholinergic tone and controls thermo-regulated signaling at the neuromuscular junction" reports the surprising finding that TIGER knockout mice are resistant to cold challenge because of alterations to cholinergic neurons. The study is well executed, clearly written and describes a logical progression of the discovery that TIGER in cholinergic neurons is responsible for the resistance to cold in whole body knockout mice. Sophisticated metabolite profiling is used to determine cholinergic neurons and in vitro SH-SY5Y neurons have increased glycolysis in the absence of TIGER, and further that this loss of TIGER in cholinergic neurons has important physiological consequences for muscle function and shivering thermogenesis.

I have only a few marginal comments that the author may wish to consider.

Comment on which cholinergic neurons are responsible for the cold resistance in TIGER knockout mice, peripheral or central?

Central NFkB has an important role in neuronal dendrite growth. Can you rule out NFkB signaling as a contributing factor to enhanced cold tolerance in the TIGER knockout mice?

There are far too many "data not shown" comments in the paper; please show the data in the supplement (lines 99, 113, 119, 159, 168 and 221).

While acetylcholine production relies heavily on acetyl-CoA availability, another important rate limiting step is the availability of choline controlled by the choline pump on the postsynaptic terminal after acetylcholine is degraded in the junction. This could be the rate limiting step during shivering thermogenesis or exercise. Did you assess expression of this transporter in the mouse models?

TIGER cholinergic neuron specific knockout and whole-body knockout should also have differences in exercise capacity and grip strength. Did the authors assess these physical parameters?

Was there a difference in muscle weights, fiber type or myotube diameter in the TIGER knockout mouse models?*Reviewer #3 (Recommendations for the authors):*

Was media lactate enrichment examined?

Provide a more complete description of the H218O study. Was the enrichment data normalized to body water enrichment? If not, please point that out and provide justification (e.g. were the mice body weight and composition matched?). Why was water administered by gavage and IP? The IP water injection is quite a large volume. Was it administered as a saline solution? If not, please justify (i.e. can a pain response be excluded?). How was the timing of the tissue collection chosen? Is this timing indicative steady state ATP turnover, or is it closer to an initial rate?

---

## [Author Response]

Essential revisions:1) Report body weights and body composition to exclude contributions from increased muscle mass. Metabolic cage data would eliminate alterations in physical activity, food consumption and basal energy expenditure as factors that protect from cold exposure.

We now include lean/fat mass, activity, food consumption and basal energy expenditure in new Figure 1—figure supplement 1 D, F, G and H. We also include skeletal muscle histology in new Figure 1—figure supplement 1 E.

2) Either provide data to support increased metabolic rates in the cell model (i.e. VO2, kinetic enrichment data, etc.) or reword the few discussion points that imply increased acetyl-CoA rates of production from glycolysis drive the cholinergic effect. As a side point, acetyl-CoA concentration could be measured in the cell model.

We have included the oxygen consumption rate (new Figure 6—figure supplement E) and have determined acetyl-CoA and acetyl-carnitine levels presented in new Figure 6—figure supplement C and D.

Reviewer #1 (Recommendations for the authors):Can the authors explained what they mean by "since UCP1 protein levels lag mRNA expression there was no detectable UCP1 protein" in iWAT. Could it be that there just isn't detectable amount of protein there?

The statement “since UCP1 protein levels lag mRNA…..etc..” was intended exactly as you indicated, that we could not measure any change in UCP1 protein although at this time point there was a measurable increase in UCP1 mRNA. This unclear statement has now been removed from the text.

Authors indicate that both UKO and TKO mice were "both with C57BL/6J background (line 128) but methods indicate that UKO were on C57BL/6N background.

The background strain for the UKO mice was C57BL/6J

Loading controls are missing for Figure 1G.

The loading control for Figure 1 G is now included.

Figure 4A's resolution is too low to be convincing.

We have increased the resolution of Figure 4 A. The difficulty in visualization is the size of the image due to space limitations. If you enlarge the image the differences are readily apparent.

Line 149: Figure S1G rather than 1G.

Figure S1G rather the 1 G has been corrected.

Statistical analysis: please describe two-way ANOVA multiple comparisons (what post-hoc tests were conducted when there was significant interactions/main effects?).

We have provided a detailed description of the statistical tests in the Method details section. We also have modified the description of statistical test in the figure legends.

Reviewer #2 (Recommendations for the authors):This interesting manuscript entitled "Neuronal glucose metabolism sets cholinergic tone and controls thermo-regulated signaling at the neuromuscular junction" reports the surprising finding that TIGER knockout mice are resistant to cold challenge because of alterations to cholinergic neurons. The study is well executed, clearly written and describes a logical progression of the discovery that TIGER in cholinergic neurons is responsible for the resistance to cold in whole body knockout mice. Sophisticated metabolite profiling is used to determine cholinergic neurons and in vitro SH-SY5Y neurons have increased glycolysis in the absence of TIGER, and further that this loss of TIGER in cholinergic neurons has important physiological consequences for muscle function and shivering thermogenesis.I have only a few marginal comments that the author may wish to consider.Comment on which cholinergic neurons are responsible for the cold resistance in TIGER knockout mice, peripheral or central?

Curare only inhibits the acetylcholine receptor present on skeletal muscle although the cholinergic TIGAR knockout will be present in all cholinergic neurons (both peripheral and central). Thus, our data are consistent with a peripheral neuronal effect, but we cannot rule other potential changes in central cholinergic signaling.

Central NFkB has an important role in neuronal dendrite growth. Can you rule out NFkB signaling as a contributing factor to enhanced cold tolerance in the TIGER knockout mice?

Since TIGAR can also regulate NF-κB signaling we also cannot rule out a contributing role from this pathway in altering neuronal growth or action potentials at the present time.

There are far too many "data not shown" comments in the paper; please show the data in the supplement (lines 99, 113, 119, 159, 168 and 221).

As indicated above, we now include the previously data not shown that are pertinent to the conclusions of this study.

While acetylcholine production relies heavily on acetyl-CoA availability, another important rate limiting step is the availability of choline controlled by the choline pump on the postsynaptic terminal after acetylcholine is degraded in the junction. This could be the rate limiting step during shivering thermogenesis or exercise. Did you assess expression of this transporter in the mouse models?

As suggested, we now include immunoblots for the choline transporter (new Figure 4 H) that remains unchanged in these experimental models.

TIGER cholinergic neuron specific knockout and whole-body knockout should also have differences in exercise capacity and grip strength. Did the authors assess these physical parameters?

We have accessed grip strength and exercise capacity, but these parameters were unchanged at room temperature. Since there is no increase in cholinergic signaling at room temperature but only when the mice are subjected to cold stress likely accounts for the absence of any difference at room temperature. In addition, shivering activity and normal muscle contraction in terms of strength and running are markedly different. The former results in symmetric uniformed contractions between fibers whereas shivering results from asymmetric contractions with no net directional force. As far as we are aware, the exact neuronal firing mechanisms responsible for these differences remains unknown.

Was there a difference in muscle weights, fiber type or myotube diameter in the TIGER knockout mouse models?

Muscle mass was unchanged, and we now include H and E staining of the soleus (red) and EDL (white) skeletal muscle that demonstrate no change in muscle fiber size or morphology.

Reviewer #3 (Recommendations for the authors):Was media lactate enrichment examined?

We have not directly measured media lactate. We did perform Seahorse analyses and although the TKO cells displayed increased oxygen consumption rate (OCR) there was no difference in extracellular acidification rate (ECAR). This could be argued against there being an increase in glycolysis, but lactate formation is a measure of anaerobic glycolysis, necessary for regeneration of NAD^+^ for glycolysis maintenance therein, or spillover of pyruvate flux that outweighs PDH activity under normal conditions. In our case, we see increased (mitochondrial) PDH flux, and the increased glucose flux entering the pyruvate pool winds up being metabolized in the TCA cycle rather than converting to lactate. This is particularly true for neurons that uptake lactate and then convert it to pyruvate for energy production in the mitochondria.

Provide a more complete description of the H218O study. Was the enrichment data normalized to body water enrichment? If not, please point that out and provide justification (e.g. were the mice body weight and composition matched?). Why was water administered by gavage and IP? The IP water injection is quite a large volume. Was it administered as a saline solution? If not, please justify (i.e. can a pain response be excluded?). How was the timing of the tissue collection chosen? Is this timing indicative steady state ATP turnover, or is it closer to an initial rate?

A more detailed description of the ^18^O water labeling and normalization to body water enrichment is now included in the Method details section.

Mice body weights and composition were the same as now shown in new Figure 1—figure supplement 1.

The IP was for a bolus injection to immediately raise the ^18^O water enrichment. The gavage was to sustain the enrichment to reach steady state.

The saline ^18^O-water (0.9% NaCl in ^18^O water) was used for IP injection. We cannot exclude some discomfort due to the injection. Since the control and TKO mice were treated identically, the differences in labeling are not attributed to a differential pain response.

The timing was chosen so that labeling reach steady-state that is now provided in the Methods detailed section.